# Categorical representation from sound and sight in the ventral occipito-temporal cortex of sighted and blind

Stefania Mattioni[1]*, Mohamed Rezk[1,2], Ceren Battal[1,2], Roberto Bottini[2], Karen E Cuculiza Mendoza[2], Nikolaas N Oosterhof[2], Olivier Collignon[1]*

[1]Institute of research in Psychology (IPSY) & Institute of Neuroscience (IoNS) - University of Louvain (UCLouvain), Louvain-la-Neuve, Belgium; [2]Centre for Mind/Brain Sciences, University of Trento, Trento, Italy

**Abstract** Is vision necessary for the development of the categorical organization of the Ventral Occipito-Temporal Cortex (VOTC)? We used fMRI to characterize VOTC responses to eight categories presented acoustically in sighted and early blind individuals, and visually in a separate sighted group. We observed that VOTC reliably encodes sound categories in sighted and blind people using a representational structure and connectivity partially similar to the one found in vision. Sound categories were, however, more reliably encoded in the blind than the sighted group, using a representational format closer to the one found in vision. Crucially, VOTC in blind represents the categorical membership of sounds rather than their acoustic features. Our results suggest that sounds trigger categorical responses in the VOTC of congenitally blind and sighted people that partially match the topography and functional profile of the visual response, despite qualitative nuances in the categorical organization of VOTC between modalities and groups.

*For correspondence:
stefania.mattioni@uclouvain.be
(SM);
olivier.collignon@uclouvain.be
(OC)

Competing interests: The authors declare that no competing interests exist.

## Introduction

The study of sensory deprived individuals represents a unique model system to test how sensory experience interacts with intrinsic biological constraints to shape the functional organization of the brain. One of the most striking demonstrations of experience-dependent plasticity comes from studies of blind individuals showing that the occipital cortex (traditionally considered as visual) massively extends its response repertoire to non-visual inputs (*Neville and Bavelier, 2002*; *Sadato et al., 1998*).

But what are the mechanisms guiding this process of brain reorganization? It was suggested that the occipital cortex of people born blind is repurposed toward new functions that are distant from the typical tuning of these regions for vision (*Bedny, 2017*). In fact, the functional organization of occipital regions has been thought to develop based on innate protomaps implementing a computational bias for low-level visual features including retinal eccentricity bias (*Malach et al., 2002*), orientation content (*Rice et al., 2014*), spatial frequency content (*Rajimehr et al., 2011*) and the average curvilinearity/rectilinearity of stimuli (*Nasr et al., 2014*). This proto-organization would serve as low-level visual biases scaffolding experience-dependent domain specialization (*Arcaro and Livingstone, 2017*; *Gomez et al., 2019*). Consequently, in absence of visual experience, the functional organization of the occipital cortex could not develop according to this visual proto-organization and those regions may therefore switch their functional tuning toward distant computations (*Bedny, 2017*).

In striking contrast with this view, several studies suggested that the occipital cortex of congenitally blind people maintains a division of computational labor somewhat similar to the one characterizing the sighted brain (*Amedi et al., 2010*; *Dormal and Collignon, 2011*; *Ricciardi et al., 2007*). Perhaps, the most striking demonstration that the occipital cortex of blind people develops a similar

**eLife digest** The world is full of rich and dynamic visual information. To avoid information overload, the human brain groups inputs into categories such as faces, houses, or tools. A part of the brain called the ventral occipito-temporal cortex (VOTC) helps categorize visual information. Specific parts of the VOTC prefer different types of visual input; for example, one part may tend to respond more to faces, whilst another may prefer houses. However, it is not clear how the VOTC characterizes information.

One idea is that similarities between certain types of visual information may drive how information is organized in the VOTC. For example, looking at faces requires using central vision, while looking at houses requires using peripheral vision. Furthermore, all faces have a roundish shape while houses tend to have a more rectangular shape. Another possibility, however, is that the categorization of different inputs cannot be explained just by vision, and is also be driven by higher-level aspects of each category. For instance, how humans use or interact with something may also influence how an input is categorized. If categories are established depending (at least partially) on these higher-level aspects, rather than purely through visual likeness, it is likely that the VOTC would respond similarly to both sounds and images representing these categories.

Now, Mattioni et al. have tested how individuals with and without sight respond to eight different categories of information to find out whether or not categorization is driven purely by visual likeness. Each category was presented to participants using sounds while measuring their brain activity. In addition, a group of participants who could see were also presented with the categories visually. Mattioni et al. then compared what happened in the VOTC of the three groups – sighted people presented with sounds, blind people presented with sounds, and sighted people presented with images – in response to each category.

The experiment revealed that the VOTC organizes both auditory and visual information in a similar way. However, there were more similarities between the way blind people categorized auditory information and how sighted people categorized visual information than between how sighted people categorized each type of input. Mattioni et al. also found that the region of the VOTC that responds to inanimate objects massively overlapped across the three groups, whereas the part of the VOTC that responds to living things was more variable.

These findings suggest that the way that the VOTC organizes information is, at least partly, independent from vision. The experiments also provide some information about how the brain reorganizes in people who are born blind. Further studies may reveal how differences in the VOTC of people with and without sight affect regions typically associated with auditory categorization, and potentially explain how the brain reorganizes in people who become blind later in life.

coding structure and topography as the one typically observed in sighted people comes from studies exploring the response properties of the ventral occipito-temporal cortex (VOTC). In sighted individuals, lesion and neuroimaging studies have demonstrated that VOTC shows a medial to lateral segregation in response to living and non-living visual stimuli, respectively, and that some specific regions respond preferentially to visual objects of specific categories like the fusiform face area (FFA; *Kanwisher et al., 1997*; *Tong et al., 2000*), the extrastriate body area (EBA; *Downing et al., 2001*) or the parahippocampal place area (PPA; *Epstein and Kanwisher, 1998*). Interestingly, In early blind people, the functional preference for words (*Reich et al., 2011*) or letters (*Striem-Amit et al., 2012*), motion (*Dormal et al., 2016*; *Poirier et al., 2004*), places (*He et al., 2013*; *Wolbers et al., 2011*), bodies (*Kitada et al., 2014*; *Striem-Amit and Amedi, 2014*), tools (*Peelen et al., 2013*) and shapes (*Amedi et al., 2007*) partially overlaps with similar categorical responses in sighted people when processing visual inputs.

Distributed multivariate pattern analyses (*Haxby et al., 2001*) have also supported the idea that the large-scale categorical layout in VOTC shares similarities between sighted and blind people (*Handjaras et al., 2016*; *van den Hurk et al., 2017*; *Peelen et al., 2014*; *Wang et al., 2015*). For example, it was shown that the tactile exploration of different manufactured objects (shoes and bottles) elicits distributed activity in VOTC of blind people similar to the one observed in sighted people in vision (*Pietrini et al., 2004*). A recent study demonstrated that the response patterns elicited by

sounds of four different categories in the VOTC of blind people could successfully predict the categorical response to images of the same categories in the VOTC of sighted controls, suggesting overlapping distributed categorical responses in sighted for vision and in blind for sounds (*van den Hurk et al., 2017*). All together, these studies suggest that there is more to the development of the categorical response of VOTC than meets the eye (*Collignon et al., 2012*).

However, these researches leave several important questions unanswered. If a spatial overlap exists between the sighted processing visual inputs and the blind processing non-visual material, whether VOTC represents similar informational content in both groups remains unknown. It is possible, for instance, that the overlap in categorical responses between groups comes from the fact that VOTC represents visual attributes in the sighted (*Arcaro and Livingstone, 2017*; *Gomez et al., 2019*) and acoustic attributes in the blind due to crossmodal plasticity (*Bavelier and Neville, 2002*). Indeed, several studies involving congenitally blind have shown that their occipital cortex may represent acoustic features – for instance frequencies (*Huber et al., 2019*; *Watkins et al., 2013*) – which form the basis of the development of categorical selectivity in the auditory cortex (*Moerel et al., 2012*). Such preferential responses for visual or acoustic features in the sighted and blind, respectively, may lead to overlapping patterns of activity for similar categories while implementing separate computations on the sensory inputs. Alternatively, it is possible that the VOTC of both groups code for higher-order categorical membership of stimuli presented in vision in sighted and audition in the blind, at least partial independently from low-level features of the stimuli.

Moreover, the degree of similarity between the categorical representation in sighted and in blind might differ across different categories: not all the regions in VOTC seem to be affected to the same extent by the crossmodal plasticity reorganization (*Bi et al., 2016*; *Dormal et al., 2018*; *Wang et al., 2015*). This 'domain–by–modality interaction' suggests that intrinsic characteristics of objects belonging to different categories might drive this difference. However, a qualitative exploration of the structure of the categorical representation in the VOTC of blind and sighted is still missing.

Another unresolved but important question is whether sighted people also show categorical responses in VOTC to acoustic information similar to the one they show in vision. For instance, the two multivariate studies using sensory (not word) stimulation (tactile, *Pietrini et al., 2004*; auditory, *van den Hurk et al., 2017*) of various categories in sighted and blind either did not find the existence of category-related patterns of response in the ventral temporal cortex of sighted people (*Pietrini et al., 2004*) or did not report overlapping distributed response between categories presented acoustically or visually in sighted people (*van den Hurk et al., 2017*). Therefore, it remains controversial whether similar categorical responses in VOTC for visual and non-visual sensory stimuli only emerge in the absence of bottom-up visual inputs during development or whether it is an organizational property also found in the VOTC of sighted people.

Finally, it has been suggested that VOTC regions might display similar functional profile for sound and sight in sighted and blind because different portions of this region integrate specific large-scale brain networks sharing similar functional coding. However, empirical evidence supporting this mechanistic account remains scarce.

With these unsolved questions in mind, we relied on a series of complementary multivariate analyses in order to carry out a comprehensive mapping of the representational geometry underlying low-level (acoustic or visual features mapping) and categorical responses to images and sounds in the VOTC of sighted and early blind people.

All together our results demonstrate that early visual deprivation triggers an extension of the intrinsic, partially non-visual, categorical organization of VOTC, potentially supported by a connectivity bias of portions of VOTC with specific large-scale functional brain networks. However, the categorical representation of the auditory stimuli in VOTC of both blind and sighted individuals exhibits different qualitative nuances compared to the categorical organization generated by visual stimuli in sighted people.

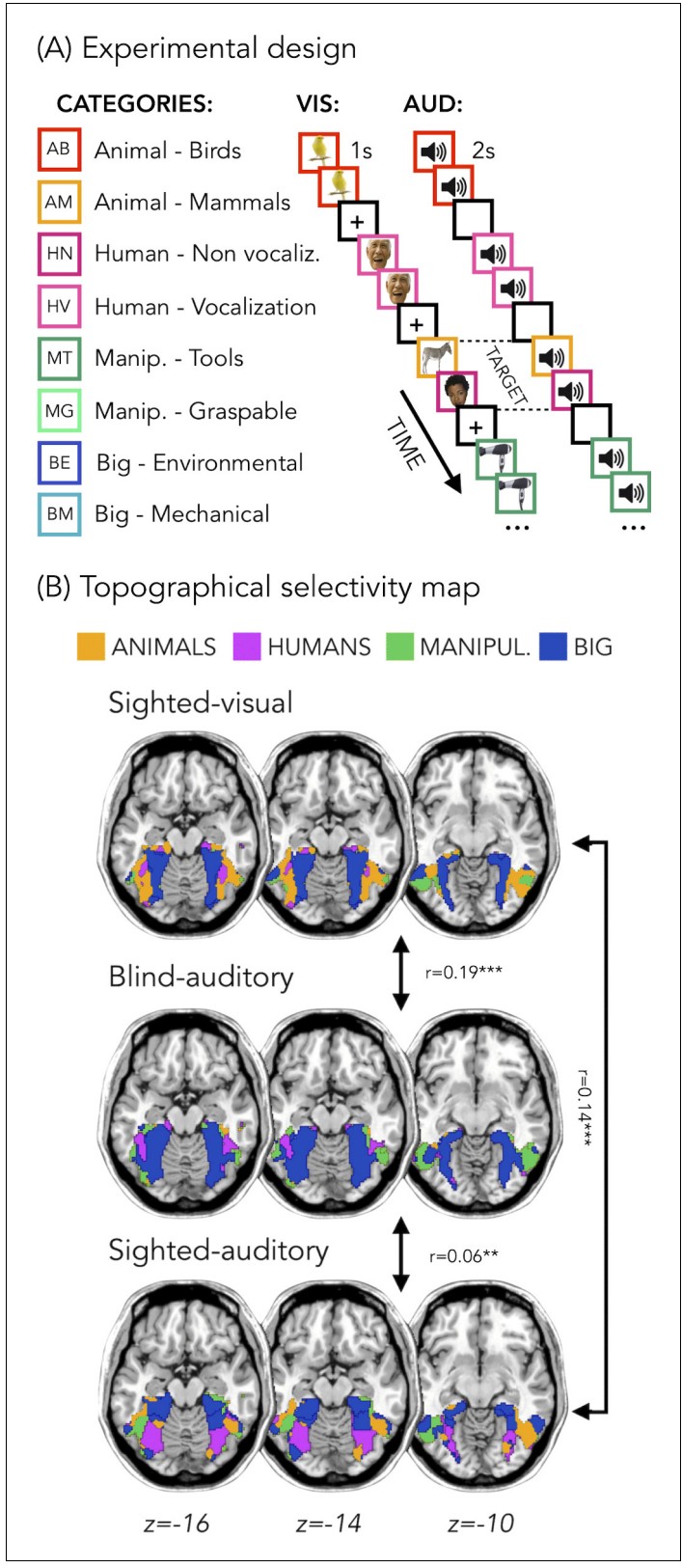

**Figure 1.** Experimental design and topographical selectivity maps. (**A**) Categories of stimuli and design of the visual (VIS) and auditory (AUD) fMRI experiments. (**B**) Averaged untresholded topographical selectivity maps for the sighted-visual (top), the blind-auditory (center) and the sighted-auditory (bottom) participants. These maps visualize the functional topography of VOTC to the main four categories in each group. These group maps are

*Figure 1 continued on next page*

*Figure 1 continued*

created for visualization purpose only since statistics are run from single subject maps (see methods). To obtain those group maps, we first averaged the β-values among participants of the same group in each voxel inside the VOTC mask for each of our 4 main conditions (animals, humans, manipulable objects and places) separately and we then assign to each voxel the condition producing the highest β-value. We decided to represent maps including the 4 main categories (instead of 8) to simplify visualization of the main effects (the correlation values are almost identical with 8 categories and those maps can be found in supplemental material).

The online version of this article includes the following figure supplement(s) for figure 1:

**Figure supplement 1.** Topographical selectivity map for 8 categories.

## Results

### Topographical selectivity map

*Figure 1B* represents the topographical selectivity maps, which show the voxel-wise preferred stimulus condition based on a winner take-all approach (for the four main categories: animals, humans, small objects and places). In the visual modality, we found the well-known functional selectivity map for visual categories (*Julian et al., 2012*; *Kanwisher, 2010*). The auditory selectivity maps of the blind subjects partially matched the visual map obtained in sighted controls during vision (r = 0.19, $p_{FDR}$ <0.001). The blind map and the visual control map are strongly correlated.

In addition, a similar selectivity map was also observed in the sighted controls using sounds. The correlation was significant both with visual map in sighted (r = 0.14, $p_{FDR}$ <0.001), and with the auditory map in blinds (r = 0.06, $p_{FDR}$ = 0.001). The correlation between EBa and SCa was significantly lower than both the correlation between SCv and EBa ($p_{FDR.}$ = 0.0003) and the correlation between SCv and SCa ($p_{FDR}$ = 0.0004). Instead, the magnitude of correlation between EBa and SCv was not significantly different from the correlation between SCa and SCv ($p_{FDR}$ = 0.233).

In *Figure 1B* we report the results on the four main categories for the simplicity of visualization, however in the supplemental material we show that the results including eight categories are almost identical (*Figure 1—figure supplement 1*).

In order to look at the consistency of the topographical representation of the categories across subjects within the same group we computed the Jaccard similarity between the topographical selectivity map of each subject and the mean topographical map of his own group. The one sample T-test revealed a significant Jaccard similarity in each category and in each group (in all cases p<0.001, after FDR correction for the 12 comparisons: three groups * four categories), highlighting a significant consistency of the topographical maps for each category between subjects belonging to the same group. We performed a repeated measures ANOVA to look at the differences between categories and groups. We obtained a significant main effect of Category ($F_{(3,138)}$=18.369; p=<0.001) and a significant interaction Group*Category ($F_{(6,138)}$=4.9; p=<0.001), instead the main effect of Group was not significant ($F_{(2,46)}$=1.83; p.17). We then run post-hoc. In SCv we did not find any difference between categories, meaning that the consistency (i.e. the Jaccard similarity) of the topographical maps for the visual stimuli in sighted was similar for each category. In the SCa group, the only two significant differences emerged between the category 'big objects and places' and the two animate categories: humans (p=0.01) and animals (p<0.008). In both cases the consistency of the topographical maps in the big object and places was significantly higher compared to the consistency in the humans and animals' categories. Finally, in the blind group only the humans and the manipulable objects categories did not show a significant difference. The animal category was, indeed, significantly lower than the humans (p=0.01), the manipulable objects (p=0.008) and the big objects and places (p=0.002). Both, the human and the manipulable objects categories were significantly lower compared to the big objects and places category (p=0.004 and p=0.007). Finally, when we look at the difference between the three groups for each category, the main differences emerged from the animals and the big objects and places categories. The animals' category showed a significantly lower Jaccard similarity within the EBa group compared to both SCa (p=0.008) and SCv (p=0.001) groups while the similarity of big objects and places category was significantly higher in EBa compared to SCv (p=0.038).

In addition, we wanted to explore the similarity and differences among the topographical representations of our categories when they were presented visually compared to when they were presented acoustically, both in sighted and in blind. To do that, we computed the Jaccard similarity index for each category, between the topographical map of each blind and sighted subject in the auditory experiment and the averaged topographical selectivity map of the sighted in the visual experiment (see *Figure 2C* for the results). The one-sample T-tests revealed a significant similarity between EBa and SCv and between SCa and SCv in each category (pFDR <0.001 in all cases). The repeated measures ANOVA highlighted a significant main effect of Category ($F_{(2.2, 69.8)}$ = 31.17, p<0.001). In both groups' comparisons the Jaccard similarity was higher between the big objects and places category compared to the other three categories (animal: p<0.001; human: p<0.001; manipulable: p<0.001).

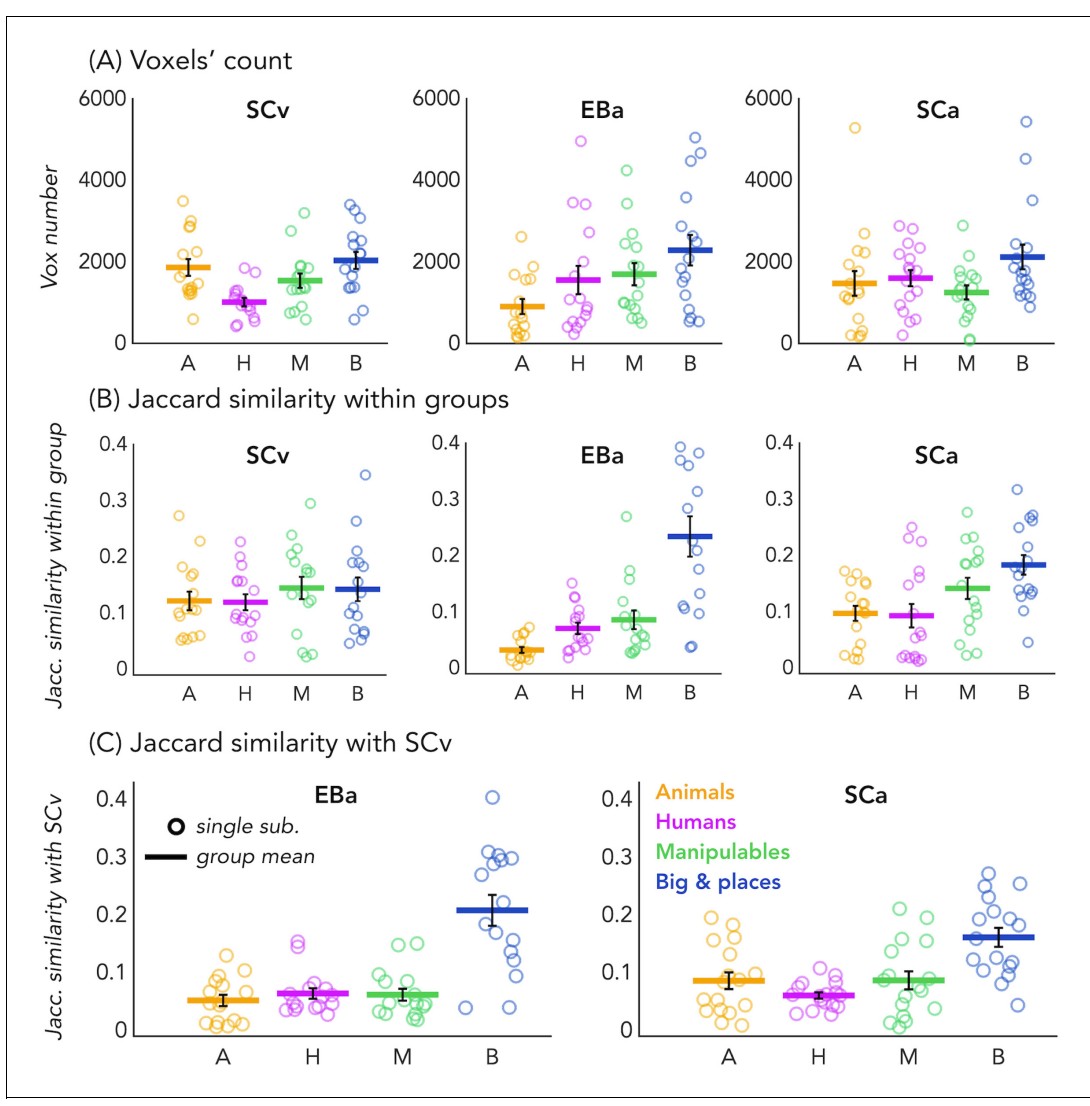

**Figure 2.** Voxels' count and Jaccard analyses. (A) Number of selective voxels for each category in each group, within VOTC. Each circle represents one subject, the colored horizontal lines represent the group average and the vertical black bars are the standard error of the mean across subjects. (B) The Jaccard similarities values within each group, for each of the four categories. (C) The Jaccard similarity indices between the EBa and the SCv groups (left side) and the Jaccard similarity indices between the SCa and the SCv groups (right side).

The online version of this article includes the following figure supplement(s) for figure 2:

**Figure supplement 1.** Voxels' count and Jaccard analyses for eight categories.

No difference emerged between groups, suggesting comparable level of similarity of the auditory topographical maps (both in blind and in sighted) with the visual topographical map in sighted participants.

Finally, since the degree of overlap highlighted by the Jaccard similarity values might be driven by the number of voxels selective for each category, we looked at the number of voxels showing the preference for each category in each group. The one sample T-test revealed a significant number of selective voxels in each category and in each group (in all cases p<0.001, after FDR correction for the 12 comparisons: 3 groups * 4 categories). We performed a repeated measure ANOVA to look at the differences between categories and groups. In SCv we found that a smaller number of voxels shows selectivity for the human category compared to the others (human vs animal: $t(15)=-3.27$; $pFDR = 0.03$; human vs manipulable: $t(15)=2.60$; $pFDR = 0.08$; human vs big: $t(15)=-4.16$; $pFDR = 0.01$). In EBa, instead, there is a lower number of voxels preferring animals compare to non-living categories (animals vs manipulable: $t(15)=-2.47$; $pFDR = 0.09$; animals vs big: $t(15)=-3.22$; $pFDR = 0.03$). Finally, in SCa the number of voxels selective for the big and place category was significantly higher than the number of voxels selective for the manipulable category ($t(16)=3.27$; $pFDR = 0.03$). Importantly, when we look at the difference between the 3 groups for each category, the main difference emerged from the animal category. In this category, the ANOVA revealed a main effect of group ($F(2,46)=3.91$; $p=0.03$). The post hoc comparisons revealed that this difference was mainly driven by the reduced number of voxels selective for the animal category in EBa compared to SCv ($p=0.02$).

For the results of the same analysis on the 8 different categories see also *Figure 1—figure supplement 1* and *Figure 2—figure supplement 1*.

## Binary MVP classification

*Figure 3B* represents the results from the average binary classification analyses for each group and every ROI (FDR corrected). In SCv and in EBa the averaged decoding accuracy was significantly higher than chance level in both EVC (SCv: DA = 69%; $t_{(15)}=6.69$, $p_{FDR}$ <0.00001; EBa: DA = 55%; $t_{(15)}=4.48$, $p_{FDR} = 0.0006$) and VOTC (SCv: DA = 71%; $t_{(15)}=7.37$, $p_{FDR}$ <0.00001; EBa: DA = 57%; $t_{(15)}=8.00$, $p_{FDR}$ <0.0001). In the SCa the averaged decoding accuracy was significantly higher than the chance level in VOTC (DA = 54%; $t_{(16)}=4.32$, $p_{FDR} = 0.0006$) but not in EVC (DA = 51%; $t_{(16)}=1.70$, $p_{FDR} = 0.11$). Moreover, independent sample t-tests revealed higher decoding accuracy values in EBa when compared to SCa in both EVC ($t_{(31)}=2.52$, $p_{FDR} = 0.017$) and VOTC ($t_{(31)}=2.08$, $p_{FDR} = 0.046$).

Results from each binary classification analysis (n = 28) for each group are represented in *Figure 4* panel A1 for EVC and in panel B1 for VOTC. The p-values for each t-test is reported in the SI-table 3.

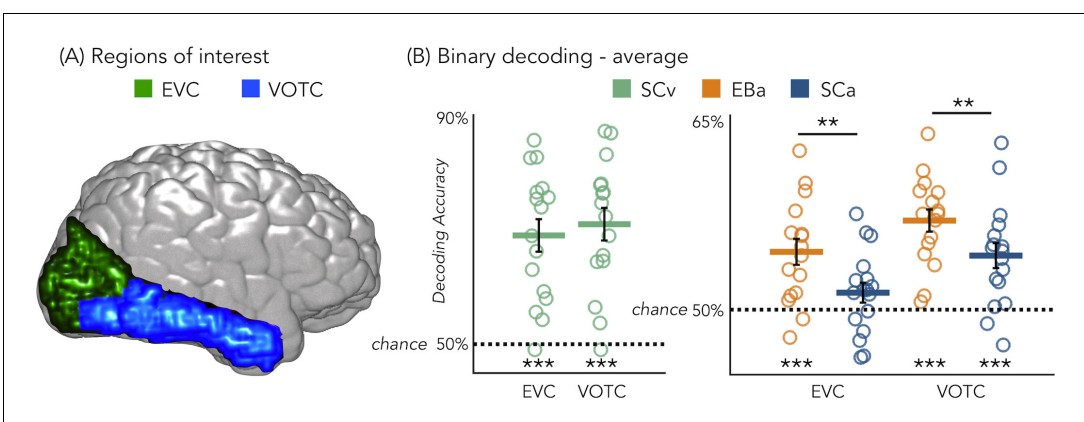

**Figure 3.** Regions of interest and classification results. (**A**) Representation of the 2 ROIs in one representative subject's brain; (**B**) Binary decoding averaged results in early visual cortex (EVC) and ventral occipito-temporal cortex (VOTC) for visual stimuli in sighted (green), auditory stimuli in blind (orange) and auditory stimuli in sighted (blue). ***p<0.001, **p<0.05.

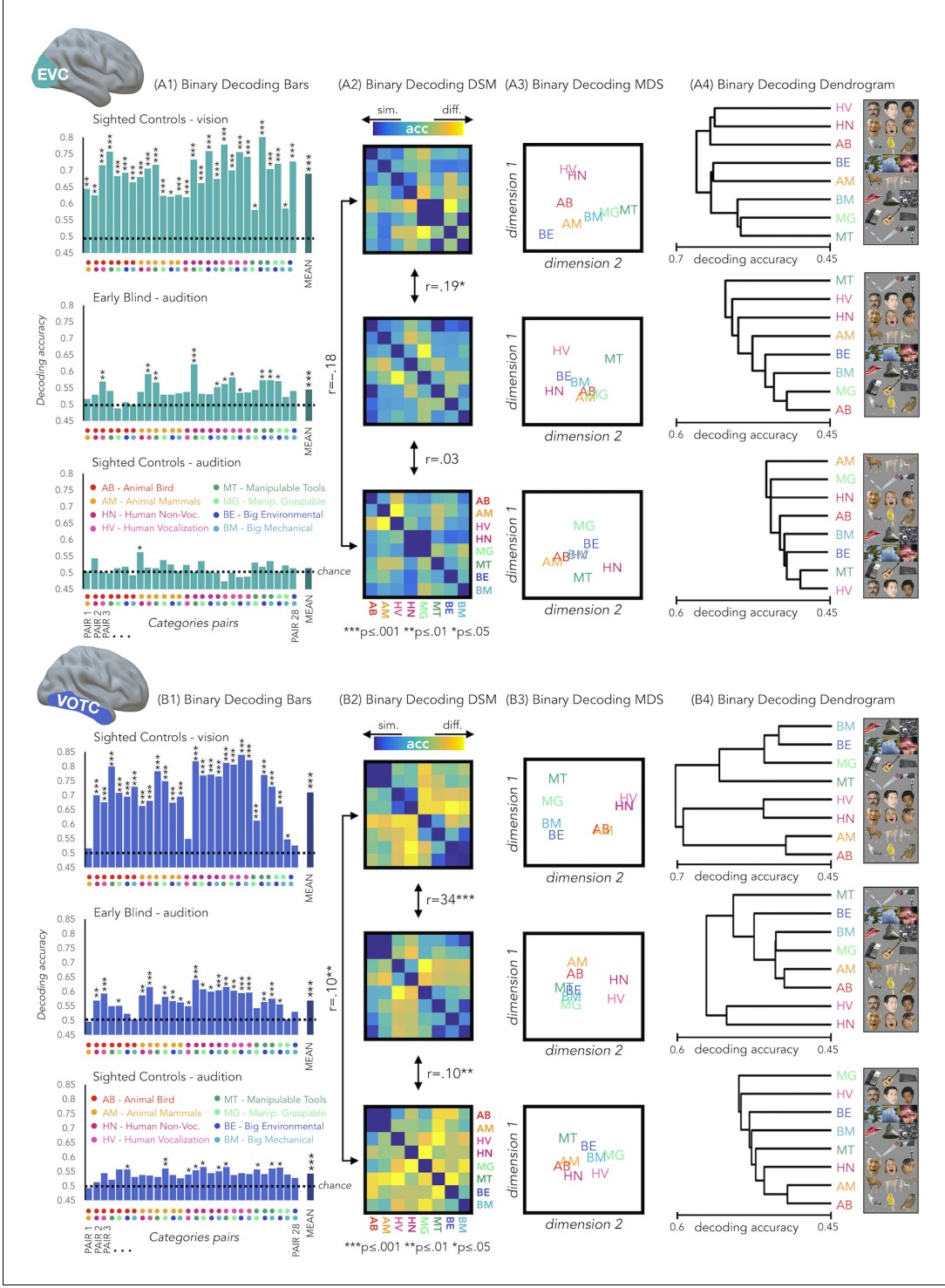

**Figure 4.** EVC and VOTC functional profiles. (**A1** and **B1**) Binary decoding bar plots. For each group (SCv: top; EBa: center; SCa: bottom) the decoding accuracy from the 28 binary decoding analyses are represented. Each column represents the decoding accuracy value coming from the classification analysis between 2 categories. The 2 dots under each column represent the 2 categories. (**A2** and **B2**) The 28 decoding accuracy values are represented in the form of a dissimilarity matrix. Each column and each row of the matrix represent one category. In each square there is the accuracy value coming from the classification analysis of 2 categories. Blue color means low decoding accuracy values and yellow color means high decoding accuracy values. (**A3** and **B3**) Binary decoding multidimensional scaling (MDS). The categories have been arranged such that their pairwise distances

*Figure 4 continued on next page*

*Figure 4 continued*

approximately reflect response pattern similarities (dissimilarity measure: accuracy values). Categories placed close together were based on low decoding accuracy values (similar response patterns). Categories arranged far apart generated high decoding accuracy values (different response patterns). The arrangement is unsupervised: it does not presuppose any categorical structure (*Kriegeskorte et al., 2008b*). (A4 and B4) Binary decoding dendrogram. We performed hierarchical cluster analysis (based on the accuracy values) to assess if EVC (A4) and VOTC (B4) response patterns form clusters corresponding to natural categories in the 3 groups (SCv: top; EBa: center; SCa: bottom).

## RSA: Correlation between the neural dissimilarity matrices of the 3 groups

We used the accuracy values from the binary classifications to build neural dissimilarity matrices for each subject in EVC (*Figure 4* - Panel A2) and in VOTC (*Figure 4* - Panel B2). Then, in every ROI we computed the correlations between DSMs for each groups' pair (i.e. SCv-EBa; SCv-SCa; EBa-SCa).

In EVC, the permutation test revealed a significant positive correlation only between the SCv and the EBa DSMs (mean r = 0.19; $p_{FDR}$ = 0.0002), whereas negative correlation emerged from the correlation between the SCv and SCa DSMs (mean r = –.18; $p_{FDR}$ <0.001) and the correlation between the SCa and the EBa DSMs (mean r = –0.03; $p_{FDR}$ = 0.18). Moreover, the correlation between SCv and EBa was significantly higher compared to both the correlation between SCv and SCa (mean corr. diff = 0.38; $p_{FDR}$ = 0.008) and the correlation between SCa and EBa (mean corr. diff = 0.22; $p_{FDR}$ = 0.008).

In VOTC, we observed a significant correlation for all the groups' pairs: SCv and EBa (r = 0.34; $p_{FDR}$ = 0.0002), SCv and SCa (r = 0.1; $p_{FDR}$ = 0.002), SCa and EBa (r = 0.1; $p_{FDR}$ = 0.002). Moreover, the correlation between SCv and EBa was significantly higher compared to both the correlation between SCv and SCa (corr. diff = 0.25; $p_{FDR}$ = 0.008) and the correlation between SCa and EBa (corr. diff = 0.25; $p_{FDR}$ = 0.008).

## Hierarchical clustering analysis on the brain categorical representation

We implemented this analysis to go beyond the magnitude of correlation values and to qualitatively explore the representational structure of our ROIs in the 3 groups. Using this analysis, we can see which categories are clustered together in the brain representations according to a specified n number of clusters (see *Figure 5* for detailed results). In VOTC the most striking results are related to the way the animal category is represented in the EBa group compared to the SCv. In fact, when the hierarchical clustering was stopped at 2 clusters in SCv, we observed a clear living vs non-living distinction. In EBa, instead, the division was between humans and non-humans (including all the non-living categories plus the animals). When the clusters are 3, we see in SCv a separation into (1) nonliving; (2) human; (3) animals. In EBa, instead, the animals keep in being clustered with non-living, in a way that the 3 clusters are: (1) Non-living and animals; (2) Human vocalization voices; (3) Human non-vocalization voices. In the case of the 4 clusters, in both SCv and EBa the additional 4th cluster is represented by the manipulable-tools category, while in the EBa the animals remain with the rest of the non-living categories. In the VOTC of SCa group, the structure of the categorical representation is less straightforward, despite the significant correlation with the DSMs of SCv. For example, we cannot clearly discern the distinction into living/non-living, or into humans and animals. However, there are some specific categories such as manipulable-graspable objects, human vocalizations and environmental sounds that show a segregated representation from the others. When we observe the clustering in EVC, we see that there is not a clear categorical clustering in this ROI in none of the groups, with the exception of the SCv in which the human stimuli tend to cluster together.

Finally, the clustering analysis on the behavioral data (see *Figure 5*) revealed a very similar way of clustering the categories in the 2 groups of sighted that in the 4 clusters step show exactly the same structure: (1) Animate categories including animals and humans; (2) Manipulable Objects; (3) Big mechanical objects; (4) Big Environmental category. In the EBa, instead, we find a different clustering structure with the animal and the human categories being separated.

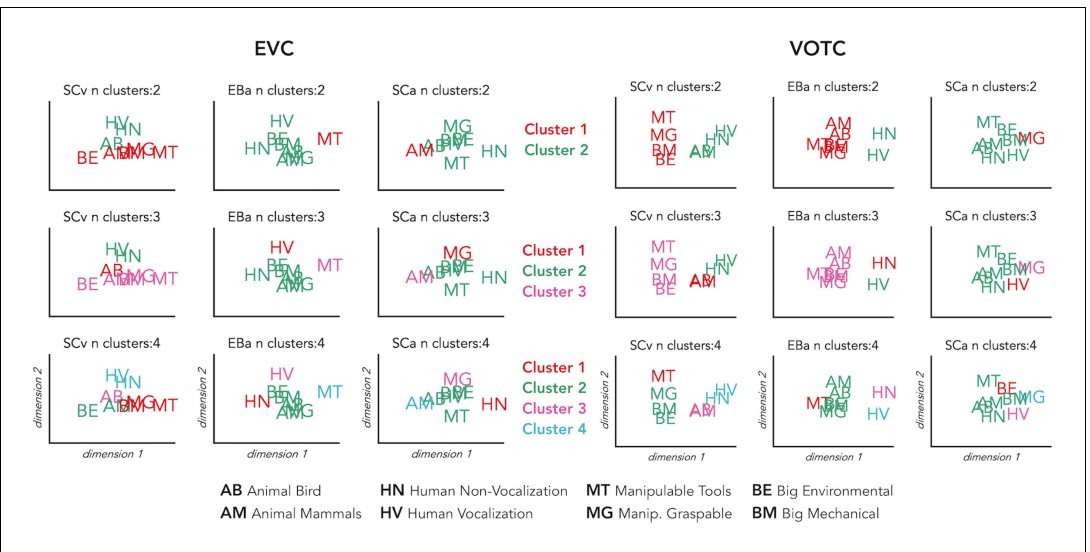

**Figure 5.** Hierarchical clustering of brain data: VOTC and EVC. Hierarchical clustering on the dissimilarity matrices extracted from EVC (left) and VOTC (right) in the three groups. The clustering was repeated three times for each DSM, stopping it at 2, 3 and 4 clusters, respectively. This allows to compare the similarities and the differences of the clusters at the different steps across the groups. In the figure each cluster is represented in a different color. The first line represents 2 clusters (green and red); the second line represents 3 clusters (green, red and pink); finally, the third line represents 4 clusters (green, red, pink and light blue).

The online version of this article includes the following figure supplement(s) for figure 5:

**Figure supplement 1.** Hierarchical clustering of the behavioral data.

## RSA: correlation with representational low-level/behavioral models

In a behavioral session following the fMRI session, we asked our participants to rate each possible pair of stimuli in the experiment they took part in (either visual or acoustic) and we built three dissimilarity matrices based on their judgments. A visual exploration of the ratings using the dissimilarity matrix visualization revealed a clustering of the stimuli into a main living/non-living distinction, with some sub-clustering such as humans, animals and objects (*Figure 6A*). The three DSMs were highly correlated (SCa/EBa: r = 0.85, p<*0.001*; SCa/SCv: r = 0.95, p<*0.001*; EBa/SCv: r = 0.89, p<*0.001*), revealing a similar way to group the stimuli across the three groups following mostly a categorical strategy to classify the stimuli. Based on this observation, we used the behavioral matrices as a categorical/high-level model to contrast with the low-level models built on the physical properties of the stimuli (HmaxC1 and pitch models, *Figure 6A*).

In order to better understand the representational content of VOTC and EVC, we computed second-order partial correlations between each ROI's DSM and our representational models (i.e. behavioral and low-level DSMs) for each participant. *Figure 6B* represents the results for the correlations between the brain DSMs in the 3 groups and the representational low-level/behavioral models DSMs (i.e. behavioral, pitch and Hmax-C1 DSMs).

The permutation test revealed that in SCv the EVC's DSM was significantly correlated with the Hmax-C1 model (mean r = 0.13, $p_{(one-tailed)FDR}$ = *0.002*) but not with the behavioral model (mean r = 0.03; $p_{(one-tailed)FDR}$ = *0.11*). Even though the correlation was numerically higher with the Hmax-C1 model than with the behavioral model, a paired samples t-test did not reveal a significant difference between the two ($t_{(15)=}$1.24, $p_{(one-tailed)FDR}$ = 0.23). The permutation test, showed that VOTC's DSM, instead, was significantly positively correlated with the behavioral model (mean r = 0.34; $p_{FDR}$ <0.001) but negative correlated with the Hmax-C1 model (r = −.07; $p_{(one-tailed)FDR}$ = *0.991*.). A paired samples t-test revealed that the difference between the correlation with the two models was significant ($t_{(15)=}$6.71, $p_{FDR}$ <0.001).

In the EBa and SCa groups, EVC's DSMs were not significantly correlated with neither the behavioral (EBa: mean r = 0.004; $p_{(one-tailed)FDR}$ = 0.47; SCa: mean r = −0.11, $p_{(one-tailed)FDR}$ = 0.98) nor the

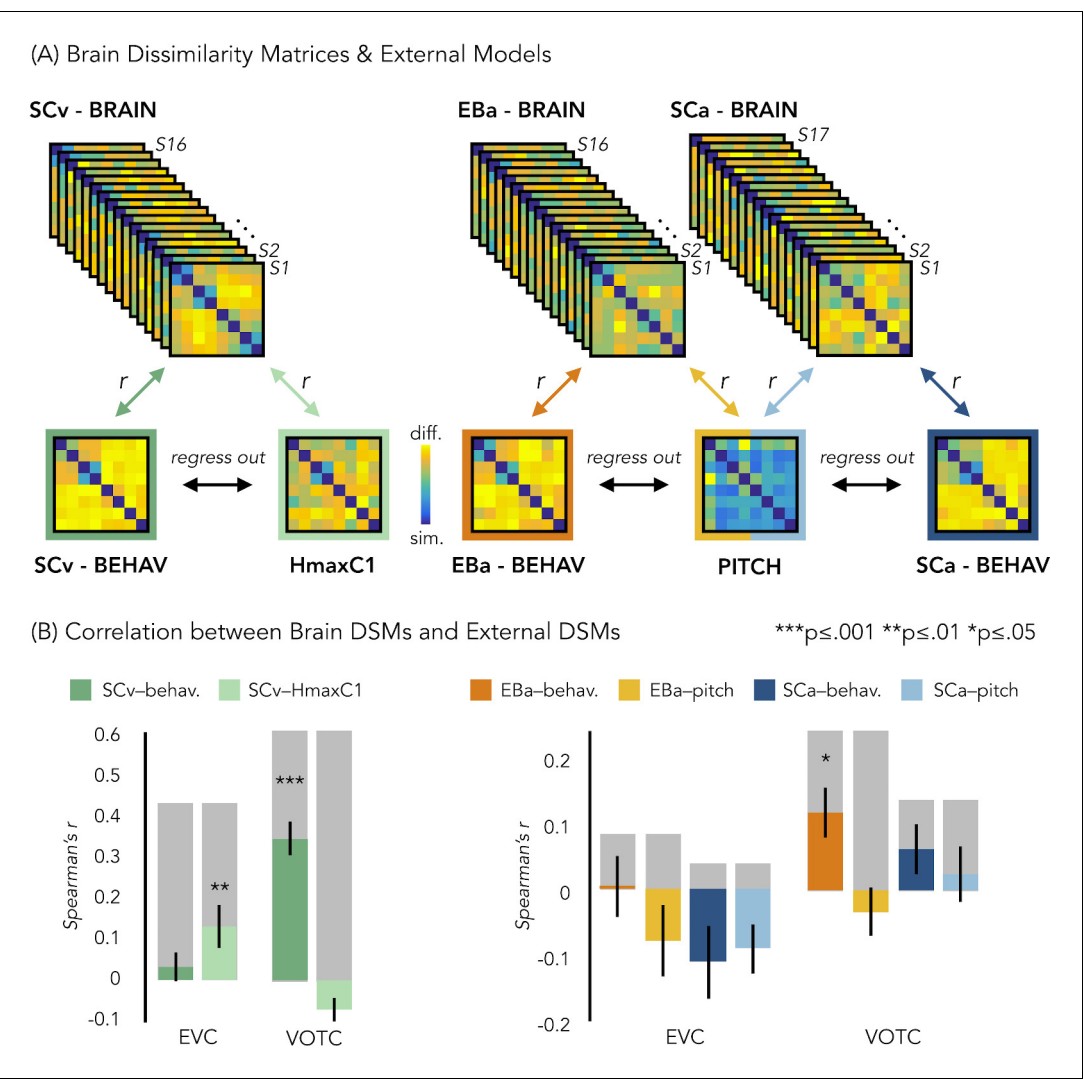

**Figure 6.** Representational similarity analysis (RSA) between brain and representational low-level/behavioral models. (**A**) In each ROI we computed the brain dissimilarity matrix (DSM) in every subject based on binary decoding of our 8 different categories. In the visual experiment (left) we computed the partial correlation between each subject's brain DSM and the behavioral DSM from the same group (SCv-Behav.) regressing out the shared correlation with the HmaxC1 model, and vice versa. In the auditory experiment (right) we computed the partial correlation between each subject's brain DSM (in both Early Blind and Sighted Controls) and the behavioral DSM from the own group (either EBa-Behav. or SCa-Behav.) regressing out the shared correlation with the pitch model, and vice versa. (**B**) Results from the Spearman's correlation between representational low-level/behavioral models and brain DSMs from both EVC and VOTC. On the left are the results from the visual experiment. Dark green: partial correlation between SCv brain DSM and behavioral model; Light green: Partial correlation between SCv brain DSM and HmaxC1 model. On the right are the results from the auditory experiment in both early blind (EBa) and sighted controls (SCa). Orange: partial correlation between EBa brain DSM and behavioral model; Yellow: Partial correlation between EBa brain DSM and pitch model. Dark blue: partial correlation between SCa brain DSM and behavioral model; Light blue: partial correlation between SCa brain DSM and pitch model. For each ROI and group, the gray background bar represents the reliability of the correlational patterns, which provides an approximate upper bound of the observable correlations between representational low-level/behavioral models and neural data (***Bracci and Op de Beeck, 2016***; ***Nili et al., 2014***). Error bars indicate SEM. ***p<0.001, **p<.005, *p<0.05. P values are FDR corrected.

The online version of this article includes the following figure supplement(s) for figure 6:

**Figure supplement 1.** Auditory model selection.

pitch model (EBa: mean r = –0.08; $p_{(one-tailed)FDR}$ = 0.94; SCa: mean r = –0.09, $p_{(one-tailed)FDR}$ = 0.98). In contrast, the VOTC's DSMs were significantly correlated with the behavioral model in EBa (mean r = 0.12, $p_{(one-tailed)FDR}$ = 0.02) but not in SCa (mean r = 0.06; $p_{(one-tailed)FDR}$ = 0.17). Finally the VOTC's DSMs were not significantly correlated with the pitch model neither in EBa(mean r = –0.03; $p_{(one-tailed)FDR}$ = 0.49), nor in SCa(mean r = 0.03; $p_{(one-tailed)FDR}$ = 0.39); In addition, a 2 Groups (EBa/SCa) X 2 Models (behavioral/pitch) ANOVA in VOTC revealed a significant main effect of Model ($F_{(1,31)}$=11.37, p=0.002) and a significant interaction Group X Model ($F_{(1,31)}$=4.03, p=0.05), whereas the main effect of Group ($F_{(1,31)}$=2.38$^{-4}$, p=0.98), was non-significant. A Bonferroni post-hoc test on the main effect of Model confirmed that the correlation was significantly higher for the behavioral model compared to the pitch model (t = 3.18, p=0.003). However, the Bonferroni post-hoc test on the interaction Group*Model revealed that the difference between behavioral and pitch models was significant only in EBa (t = 3.8, p=0.004).

For completeness of results, we report here also the correlation results before regressing out the partial correlation of the behavioral/low-level models from each other. In ECV, the mean correlation with the behavioral model was: r = 0.2 in SCv, r = 0.02 in EBa and r=–0.07 in SCa. In ECV, the mean correlation with the low level/model was: r = 0.21 in SCv (HmaxC1), r=–0.09 in EBa (pitch) and r = –0.06 in SCa (pitch). In VOTC, the mean correlation with the behavioral model was: r = 0.42 in SCv, r = 0.12 in EBa and r = 0.04 in SCa. In VOTC, the mean correlation with the low level/model was: r = 0.15 in SCv (HmaxC1), r = –0.08 in EBa (pitch) and r = 0.005 in SCa (pitch).

## RSA: Inter-subjects correlation

We run this analysis to understand how variable was the brain representation in VOTC across subjects belonging either to the same group or to different groups. Since we have 3 groups, this analysis resulted in 6 different correlation values: 3 values for the 3 within group correlation conditions (SCv; EBa; SCa) and 3 values for the 3 between groups correlation conditions (i.e. SCv-EBa; SCv-SCa; EBa-SCa). Results are represented in *Figure 7*.

The permutation test revealed that the correlation between subjects' DSMs in the within group condition was significant in SCv (r = 0.42; $p_{FDR}$ <0.001) and EBa (r = 0.10; $p_{FDR}$ <0.001), whereas it was not significant in SCa (r = –.03; $p_{FDR}$ = 0.98). Moreover, the correlation between subjects' DSMs was significant in all the three between groups conditions (SCv-EBa: r = 0.17, $p_{FDR}$ <0.001; SCv-SCa: r = 0.04, $p_{FDR}$ = 0.002; EBa-SCa: r = 0.02; $p_{FDR}$ = 0.04). When we ranked the correlations values (*Figure 7*) we observed that the highest inter-subject correlation is the within SCv group condition, which was significantly higher compared to all the other five conditions. It was followed by inter-subject correlation between SCv and EBa group and the within EBa group correlation. Interestingly, both the between groups SCv-EBa and the within group EBa correlations were significantly higher compared to the last 3 inter-subjects correlation's values (between SCv-SCa; between EBa-SCa; within SCa).

## Representational connectivity analysis

*Figure 8* represents the results from the representational connectivity analysis in VOTC. The permutation analysis highlighted that the representational connectivity profile of VOTC with the rest of the brain is significantly correlated between all pairs of groups (SCv-EBa: mean r = 0.18, $p_{FDR}$ = 0.001; SCv-SCa: mean r = 0.14, $p_{FDR}$ <0.001; EBa-SCa: mean r = 0.16, $p_{FDR}$ <0.001), and with no difference between groups' pairs.

We performed the same analysis also in EVC. In this case the permutation analysis revealed a significant correlation only between the representational connectivity profile of the two groups of sighted: SCv and SCa (mean r = 0.17, $p_{FDR}$ <0.001), whereas the correlation between the EBa was not significant neither with SCv (mean r = 0.06, $p_{FDR}$ = 0.12) nor with SCa (mean r = 0.06, $p_{FDR}$ = 0.11). Moreover, the correlation between SCv and SCa was significantly higher than both, the correlation between SCv and EBa (mean diff = 0.11, $p_{FDR}$ = 0.01) and the correlation between SCa and EBa (mean diff = 0.11, $p_{FDR}$ = 0.01).

## Discussion

In our study, we demonstrate that VOTC reliably encodes the categorical membership of sounds from eight different categories in sighted and blind people, using a topography (*Figure 1B*),

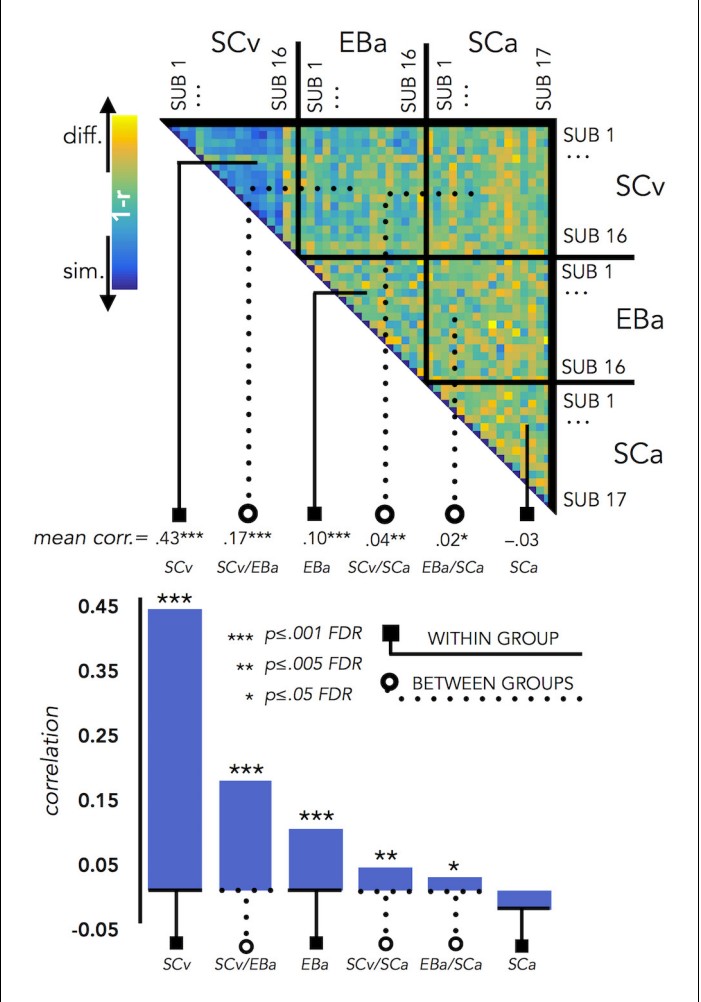

**Figure 7.** VOTC Inter-subject correlation within and between groups. Upper panel represents the correlation matrix between the VOTC brain DSM of each subject with all the other subjects (from the same group and from different groups). The mean correlation of each within- and between-groups combination is reported in the bottom panel (bar graphs). The straight line ending with a square represents the average of the correlation between subjects from the same group (i.e. within groups conditions: SCv, EBa, SCa), the dotted line ending with the circle represents the average of the correlation between subjects from different groups (i.e. between groups conditions: SCv-EBa/SCv-SCa/EBa SCa). The mean correlations are ranked from the higher to the lower inter-subject correlation values.

representational format (*Figure 4*) and a representational connectivity profile (*Figure 8*) partially similar to the one observed in response to images of similar categories in vision.

Previous studies using linguistic stimuli had already suggested that VOTC may actually represent categorical information in a more abstracted fashion than previously thought (*Handjaras et al., 2016*; *Borghesani et al., 2016*; *Striem-Amit et al., 2018b*; *Peelen and Downing, 2017*). However, even if the use of words is very useful in the investigation of pre-existing representation of concepts (*Martin et al., 2017*), it prevents the investigation of a bottom-up perceptual processing. By contrast, in our study we used sensory-related non-linguistic stimuli (i.e. sounds) in order to investigate both the sensory (acoustic) and categorical nature of the representation implemented in VOTC. To the limit of our knowledge, only one recent study investigated the macroscopic functional organization of VOTC during categorical processing of auditory and visual stimuli in sighted and in blind individuals (*van den Hurk et al., 2017*). They found that it is possible to predict the global large-scale distributed pattern of activity generated by different categories presented visually in sighted using the pattern of activity generated by the same categories presented acoustically in early blind.

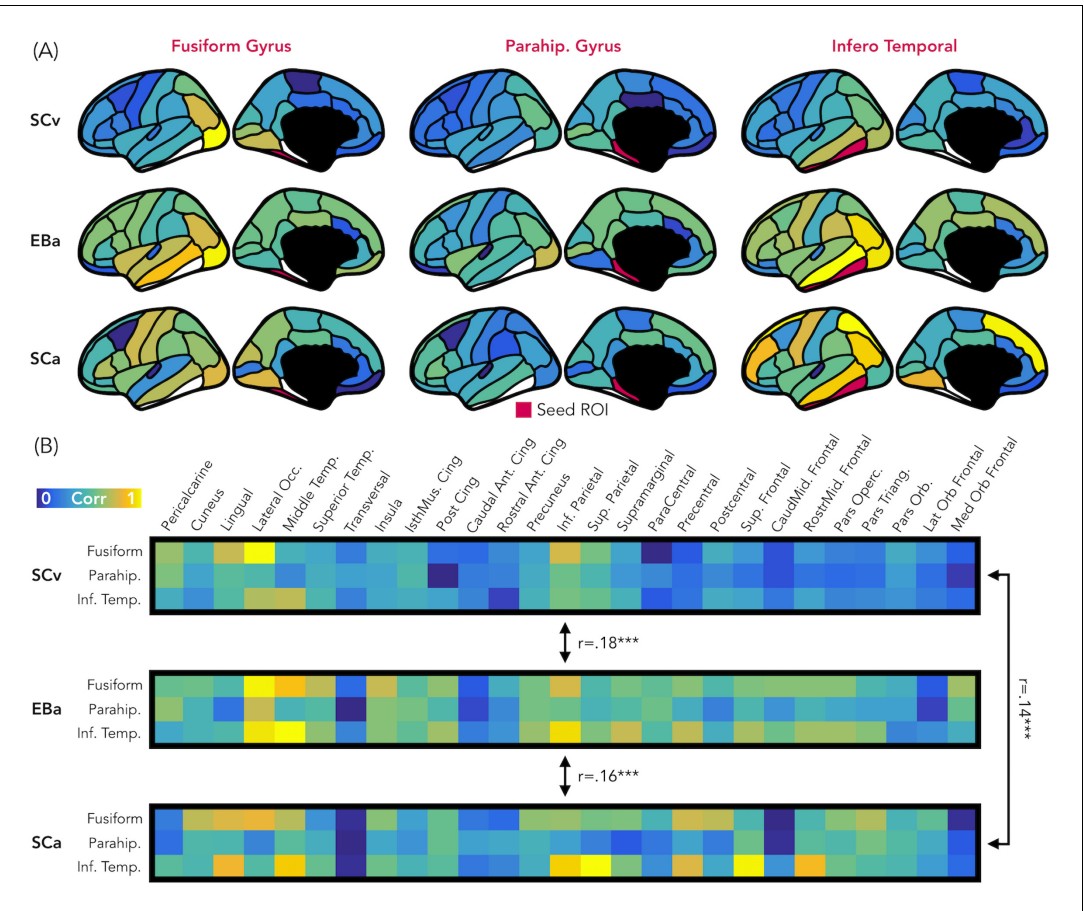

**Figure 8.** Representational connectivity. (**A**) Representation of the z-normalized correlation values between the dissimilarity matrix of the three VOTC seeds (left: Fusiform gyrus, center: Parahippocampal gyrus, Right: Infero-Temporal cortex) and the dissimilarity matrix of 27 parcels covering the rest of the cortex in the three groups (top: SCv, central: EBa, bottom: SCa). Blue color represents low correlation with the ROI seed; yellow color represents high correlation with the ROI seed. (**B**) The normalized correlation values are represented in format of one matrix for each group. This connectivity profile is correlated between groups. SCv: sighted control-vision; EBa: early blind-audition; SCa: sighted control-audition.

Relying on a different analytical stream, focusing on representational matrices extracted from pairwise decoding of our eight categories, our study confirms and extends those findings by showing that VOTC reliably encodes sound categories in blind people using a representational structure relatively similar to the one found in vision.

Our study goes beyond previous results in at least six significant ways. First, our results demonstrate that VOTC shows categorical responses to sounds in the sighted and the blind in a representational format partially similar to the one elicited by images of the same categories in sighted people (see *Figure 1B*). Observation of a similar categorical representational structure in VOTC for sounds and images in sighted people is crucial to support the idea that the intrinsic categorical organization of VOTC might be partially independent from vision even in sighted and that such intrinsic multisensory functional scaffolding may constrain the way crossmodal plasticity expresses in early blind people. Second, we observed that blind people show higher decoding accuracies and higher inter-subject consistency in the representation of auditory categories, and that the representational structure of visual categories in sighted was significantly closer to the structure of the auditory categories in blind than in sighted group (see *Figure 1B*, *Figure 3*, *Figure 4B*, *Figure 7*). This points to the idea that in absence of feed-forward visual input, VOTC increases its intrinsic representation of non-visual information. Third, VOTC shows similar large-scale representational connectivity profiles when processing images in sighted and sounds in sighted and blind people (see *Figure 8*). This

result provides strong support to the general hypothesis that the functional tuning of a region is determined by large-scale connectivity patterns with regions involved in similar coding strategies (*Behrens and Sporns, 2012*; *Mahon and Caramazza, 2011*; *Passingham et al., 2002*). Fourth, our design allowed us to investigate which dimension of our stimuli, either categorical membership or acoustic properties, may determine the response properties of VOTC to sounds. By harnessing the opportunities provided by representational similarity analysis, we demonstrate that categorical membership is the main factor that predicts the representational structure of sounds in VOTC in blind people (see *Figure 6*), rather than lower-level acoustical attributes of sounds that are at least partially at the basis of category selectivity in the temporal cortex (*Moerel et al., 2012*). These results elucidate for the first time the computational characteristics that determine the categorical response for sounds in VOTC. Fifth, we provided a qualitative exploration of the structure of the categorical representation in the VOTC. The between-groups Jaccard similarity analysis revealed a domain–by–modality interaction (see *Figure 2C*), with the big objects and places category showing an higher degree of similarity between the auditory and visual representations compared to the other categories. In addition, both, the hierarchical clustering and the within-group Jaccard similarity analysis highlighted a domain-by-sensory experience interaction (see *Figure 5* and *Figure 2B*), with the animal category represented differently in blind compared to sighted subjects (*Bi et al., 2016*; *Wang et al., 2015*). Finally, our study discloses that categorical membership is encoded in the EVC of blind people only, using a representational format that does not relate neither to the acoustic nor to the categorical structure of our stimuli, suggesting different mechanisms of reorganization in this posterior occipital region.

Different visual categories elicit distinct distributed responses in VOTC using a remarkable topographic consistency across individuals (*Julian et al., 2012*; *Kanwisher, 2010*). It was suggested that regular visual properties specific to each category like retinotopic eccentricity biases (*Gomez et al., 2019*; *Malach et al., 2002*), curvature (*Nasr et al., 2014*) or spatial frequencies (*Rajimehr et al., 2011*) could drive the development of categorical response in VOTC for visual information (*Andrews et al., 2010*; *Baldassi et al., 2013*; *Bracci et al., 2018*; *Rice et al., 2014*; see *Op de Beeck et al., 2019* for a recent review on the emergence of category selectivity in VOTC). For instance, the parahippocampal place area (PPA) and the fusiform face area (FFA) receive dominant inputs from downstream regions of the visual system with differential selectivity for high vs low spatial frequencies and peripheral vs. foveal inputs, causing them to respond differentially to place and face stimuli (*Levy et al., 2001*). These biases for specific visual attributes could be present at birth and represent a proto-organization driving the development of the categorical responses of VOTC based on experience (*Arcaro and Livingstone, 2017*; *Gomez et al., 2019*). For instance, a proto-eccentricity map is evident early in development (*Arcaro and Livingstone, 2017*) and monkeys trained early in life to discriminate different categories varying in their curvilinearity/rectilinearity develop distinct and consistent functional clusters for these categories (*Srihasam et al., 2014*). Further, adults who had intensive visual experience with Pokémon early in life demonstrate distinct distributed cortical responses to this trained visual category with a systematic location supposed to be based on retinal eccentricity (*Gomez et al., 2019*).

Although our results by no means disprove the observations that inherent visual biases can influence the development of the functional topography of high-level vision (*Gomez et al., 2019*; *Hasson et al., 2002*; *Nasr et al., 2014*), our data however suggest that category membership independently of visual attributes is also a key developmental factor that determines the consistent functional topography of the VOTC. Our study demonstrates that VOTC responds to sounds using a similar distributed functional profile to the one found in response to vision, even in case of people that have never had visual experience.

By orthogonalizing category membership and visual features of visual stimuli, previous studies reported a residual categorical effect in VOTC, highlighting how some of the variance in the neural data of VOTC might be explained by high-level categorical properties of the stimuli even when the contribution of the basic low-level features has been controlled for (*Bracci and Op de Beeck, 2016*; *Kaiser et al., 2016*; *Proklova et al., 2016*). Category-selectivity has also been observed in VOTC during semantic tasks when word stimuli were used, suggesting an involvement of the occipito-temporal cortex in the retrieval of category-specific conceptual information (*Handjaras et al., 2016*; *Borghesani et al., 2016*; *Peelen and Downing, 2017*). Moreover, previous research has shown that learning to associate semantic features (e.g., 'floats') and spatial contextual associations (e.g., 'found

in gardens') with novel objects influences VOTC representations, such that objects with contextual connections exhibited higher pattern similarity after learning in association with a reduction in pattern information about the object's visual features (*Clarke et al., 2016*).

Even if we cannot fully exclude that the processing of auditory information in the VOTC of sighted people could be the by-product of the visual imagery triggered by the non-visual stimulation (*Cichy et al., 2012*; *Kosslyn et al., 1995*; *Reddy et al., 2010*; *Slotnick et al., 2005*; *Stokes et al., 2009*), we find it unlikely. First, we purposely included two separate groups of sighted people, each one performing the experiment in one modality only, in order to minimize the influence of having heard or seen the stimuli in the other modality in the context of the experiment. Also, we used a fast event-related design that restricted the time window to build a visual image of the actual sound since the next sound was presented quickly after (*Logie, 1989*). Moreover, we would expect that visual imagery would also triggers information to be processed in posterior occipital regions (*Kosslyn et al., 1999*). Instead, we found that EVC does not discriminate the different sounds in the sighted group (*Figure 3B* and *Figure 4A*). Finally, to further test the visual imagery hypothesis, we correlated the brain representational space of EVC in SCa with low-level visual model (i.e. HmaxC1). A significant positive correlation between the two would be in support of the presence of visual imagery mechanism when sighted people hear sounds of categories. We found, instead, a non-significant negative correlation, making the visual imagery hypothesis further unlikely to explain our results.

Comparing blind and sighted individuals arguably provides the strongest evidence for the hypothesis that category-selective regions traditionally considered to be 'high-level visual regions' can develop independently of visual experience. Interestingly, we found that the decoding accuracy for the auditory categories in VOTC is significantly higher in the early blind compared to the sighted control group (*Figure 3B*). In addition, the correlation between the topographic distribution of categorical response observed in VOTC was stronger in blind versus sighted people (*Figure 1B*). Moreover, the representational structure of visual categories in sighted was significantly closer to the structure of the auditory categories in blind than in sighted (*Figure 4A2*). Finally, the representation of the auditory stimuli in VOTC is more similar between blind than between sighted subjects (*Figure 7*), showing an increased inter-subject stability of the representation in case of early visual deprivation. All together, these results not only demonstrate that a categorical organization similar to the one found in vision could emerge in VOTC in absence of visual experience, but also that such categorical response to sounds is actually enhanced and more stable in congenitally blind people.

Several studies have shown that in absence of vision, the occipital cortex enhances its response to non-visual information processing (*Collignon et al., 2012*; *Collignon et al., 2011*; *Sadato et al., 1998*). However, people debate on the mechanistic principles guiding the expression of this crossmodal plasticity. For instance, it was suggested that early visual deprivation changes the computational nature of the occipital cortex which would reorganize itself for higher-level functions, distant from the ones typically implemented for visual stimuli in the same region (*Bedny, 2017*). In contrast with this view, our results demonstrate that the expression of crossmodal plasticity, at least in VOTC (see differences in EVC below), is constrained by the inherent categorical structure endowed in this region. First, we highlighted remarkably similar functional profile of VOTC for visual and auditory stimuli in sighted and in early blind individuals (*Figure 4B*). In addition, we showed that VOTC is encoding a similar categorical dimension of the stimuli across different inputs of presentation and different visual experiences (*Figure 6B*). In support of such idea, we recently demonstrated that the involvement of right dorsal occipital region for arithmetic processing in blind people actually relates to the intrinsic 'spatial' nature of these regions, a process involved in specific arithmetic computation (e.g. subtraction but not multiplication) (*Crollen et al., 2019*). Similarly, the involvement of VOTC during 'language' as observed in previous studies (*Bedny et al., 2011*; *Burton et al., 2006*; *Kim et al., 2017*; *Lane et al., 2015*; *Röder et al., 2002*) may relate to the fact that some level of representation involved in language (e.g. semantic) can be intrinsically encoded in VOTC as supported by the current results (*Huth et al., 2016*). In fact, we suggest that VOTC regions have innate predispositions relevant to important categorical distinctions that cause category-selective patches to emerge regardless of sensory experience. Why would the 'visual' system embed representation of categories independently of their perceptual features? One argument might be that items from a particular broad category (e.g. inanimate) are so diverse that they may not share systematic perceptual features and therefore a higher-level of representation, partially abstracted from vision, might

prove important. Indeed, we gather evidence in support of an extension of the intrinsic categorical organization of VOTC that is already partially independent from vision in sighted. This finding represents an important step forward in understanding how experience and intrinsic constraints interact in shaping the functional properties of VOTC. An intriguing possibility raised by our results is that the crossmodal plasticity observed in early blind individuals may actually serve to maintain the functional homeostasis of occipital regions.

What would be the mechanism driving the preservation of the categorical organization of VOTC in case of congenital blindness? It is thought that the specific topographic location of a selective brain functions is constrained by an innate profile of functional and structural connections with extrinsic brain regions (*Passingham et al., 2002*). Since the main fiber tracts are already present in full-term human neonates (*Dubois et al., 2014*; *Dubois et al., 2016*; *Kennedy et al., 1999*; *Kostović and Judaš, 2010*; *Marín-Padilla, 2011*; *Takahashi et al., 2011*), such initial connectome may at least partly drive the functional development of a specific area. Supporting this hypothesis, the visual word form area (VWFA) in VOTC (*McCandliss et al., 2003*) shows robust and specific anatomical connectivity to EVC and to frontotemporal language networks and this connectivity fingerprint can predict the location of VWFA even before a child learn to read (*Saygin et al., 2016*). Similarly, anatomical connectivity profile can predict the location of the fusiform face area (FFA) (*Saygin et al., 2012*). In addition to intra-occipital connections, FFA has a direct structural connection with the temporal voice area (TVA) in the superior temporal sulcus (*Benetti et al., 2018*; *Blank et al., 2011*) thought to support similar computations applied on faces and voices as well as their integration (*von Kriegstein et al., 2005*). Interestingly, recent studies suggested that the maintenance of those selective structural connections between TVA and FFA explains the preferential recruitment of TVA for face processing in congenitally deaf people (*Benetti et al., 2018*; *Benetti et al., 2017*). This TVA-FFA connectivity may explain why voices preferentially map slightly more lateral to the mid-fusiform sulcus (*Figure 1B*). Similarly, sounds of big objects or natural scenes preferentially recruit more mesial VOTC regions (*Figure 1B*), overlapping with the parahippocampal place area, potentially due to the preserved pattern of structural connectivity of those regions in blind people (*Wang et al., 2017*). The existence of these innate large-scale brain connections that are specific for each region supporting separate categorical domain may provide the structural scaffolding on which crossmodal inputs capitalize to reach VOTC in both sighted and blind people, potentially through feed-back connections. Indeed, it has been shown that the main white matter tracks including those involving occipital regions are not significantly different between blind and sighted individuals (*Shimony et al., 2006*). In EB, the absence of competitive visual inputs typically coming from feed-forward inputs from EVC may actually trigger an enhanced weighting of those feed-back inter-modal connection leading to an extension of selective categorical response to sounds in VOTC, as observed in the current study. Our results provide crucial support for this 'biased connectivity' hypothesis (*Hannagan et al., 2015*; *Mahon and Caramazza, 2011*) showing that VOTC subregions are part of a large-scale functional network representing categorical information in a format that is at least partially independent from the modality of the stimuli presentation and from the visual experience.

Even though the categorical representation of VOTC appears, to a certain degree, immune to input modality and visual experience, there are also several differences emerging from the categorical representation of sight and sounds in the sighted and blind. Previous studies already suggested that intrinsic characteristics of objects belonging to different categories might drive different representations in the VOTC of the blind (*Bi et al., 2016*; *Büchel, 2003*; *Wang et al., 2015*). In line with this idea, the between-groups Jaccard similarity analysis (see *Figure 2C*) revealed a domain–by–modality interaction, with the big objects and places categories showing the highest degree of similarity between the vision and audition (both in blind and in sighted). In contrast, the lowest topographical consistency between groups was found for the animal category. We found that in the early blind group the number of voxels selective for animals is reduced compared to the other categories (see *Figure 2A*), suggesting that the animal category is under represented in the VOTC of early blind. Our hierarchical clustering analyses (see *Figure 5* and *Figure 5—figure supplement 1*) also highlight a reduced animate/inanimate division in the EBa group, with the animal and the humans categories not clustering together and the animals being represented more like tools or big objects in the EBa. Interestingly, this is the case in both the categorical representation of VOTC (*Figure 5*) and the behavioral evaluation of our stimuli made by blind individuals (*Figure 5—figure supplement*

*1*). An explanation for this effect could be the different way blind and sighted individuals might have in perceiving and interacting with animals. In fact, if we exclude pets (only 1 out of the six animals we included in this study), sighted individuals normally perceive the animacy of animals (such as bird, donkey, horse etc.) mostly throughout vision (either in real life or in pictures/movies). Blind people, instead, do normally learn the peculiar shape of each animal touching static miniature models of them. Moreover, when blind people hear the sounds of these animals without seeing them, they might combine these sounds with the rest of the environmental sounds, and this is indeed what we see in the behavioral ratings, in which only blind subjects cluster together animals and big environmental sounds. These results therefore reveal that the modality of presentation and/or the visual experience do affect the qualitative structure of the categorical representation in VOTC, and this effect is stronger for some categories (i.e. animals) compared to others (i.e. inanimate).

A different profile emerged from EVC. First, sound categories could be decoded in the EVC of EB (*Figure 3B*) but not of the SC. In addition, the representational structure of EVC for sounds correlated to the one found in vision only in EB (*Figure 4A2*). However, neither the categorical membership nor the acoustic attributes of sounds correlated with the representational structure found in the EVC of EB (*Figure 6B*). A possible explanation for this result is that the posterior part of the occipital cortex in EB is the region that distances itself the most from the native computation it typically implements (*Bi et al., 2016*; *Wang et al., 2015*). In support of this possibility, the representational connectivity profile of EVC in EBa did not show any similarity with the one of sighted (neither SCv nor SCa). Because this area has a native computation that does not easily transfer to another sensory modality (i.e. low-level vision), it may therefore rewire itself for distant functions (*Bedny, 2017*). Some studies, for instance reported an involvement of EVC in high-level linguistic or memory tasks (*Van Ackeren et al., 2018*; *Amedi et al., 2003*; *Bedny et al., 2011*). However, as demonstrated here, the categorical membership of sounds, which may be a proxy for semantic representation, does not explain the representational structure of EVC in our study. It would be interesting to investigate whether models based on linguistic properties such as word frequency or distributional statistic in language corpus (*Baroni et al., 2009*) would, at least partially, explain the enhanced information that we found in EVC of EB. However, our design does not allow us to implement this analysis because the language-statistic DSM based on our stimuli space highly correlate with categorical models. Future studies should investigate this point using a set of stimuli in which the categorical and the linguistic dimensions should be orthogonalized. A further limitation of our study is the limited number of brain regions that we investigated. Since the experimental design and analyses we implemented were a priori focused on VOTC (target region) and EVC (as a control region), it is possible that other brain areas might show either similar or different representation across modalities and groups. In particular, since the brain is a highly interconnected organ (*de Pasquale et al., 2018*), it is unlikely that early visual deprivation would affect exclusively a portion of the occipital cortex leaving the rest of the functional network unaffected. It would be of particular interest to investigate whether the reorganization of the visual cortex occurs together with changes in brain regions coding for the remaining senses, such as temporal regions typically coding for auditory stimuli (*Mattioni et al., 2018*).

## Materials and methods

### Participants

Thirty-four participants completed the auditory version of the fMRI study: 17 early blinds (EBa; 10F) and 17 sighted controls (SCa; 6F). An additional group of 16 sighted participants (SCv; 8F) performed the visual version of the fMRI experiment. All the blind participants lost sight at birth or before 4 years of age and all of them reported not having visual memories and never used vision functionally (see *Supplementary file 1*). The three groups were age (range 20–67 years, mean ± SD: 33.29 ± 10.24 for EBa subjects, respectively 23–63, 34.12 ± 8.69 for SCa subjects, and 23–51, 30.88 ± 7.24 for SCv subjects) and gender ($\chi^2$ (2,50)=1.92, p=0.38) matched. One blind subject performed only 2 out of the 5 runs in the fMRI due to claustrophobia; because of that we excluded his data. All subjects were blindfolded during the auditory task and were instructed to keep the eyes closed during the entire duration of the experiment. Participants received monetary compensation

for their participation. The ethical committee of the University of Trento approved this study (protocol 2014–007) and participants gave their informed consent before participation.

## Stimuli

We decided to use sounds and images, instead of words, because we wanted to access and model the bottom-up cascade of sensory processing starting from the low-level sensory processing up to the more conceptual level. This methodological decision was crucial in order to assess what level of sound representation is implemented in VOTC of blind and sighted individuals.

A preliminary experiment was carried out in order to select the auditory stimuli. Ten participants who did not participate in the main experiment were presented with 4 different versions of 80 acoustic stimuli from 8 different categories (human vocalization, human non-vocalization, birds, mammals, tools, graspable objects, environmental scenes, big mechanical objects). We asked the participants to name the sound and then to rate, from 1 to 7, how representative the sound was of its category. We selected only the stimuli that were recognized with at least 80% accuracy, and among those, we choose for each category the 3 most representative sounds for a total of 24 acoustical stimuli in the final set (see *Supplementary file 1*). All sounds were collected from the database *Freesound* (https://freesound.org), except for the human vocalizations that were recorded in the lab. The sounds were edited and analysed using the softwares *Audacity* (http://www.audacityteam.org) and *Praat* (http://www.fon.hum.uva.nl/praat/). Each mono-sound (44,100 Hz sampling rate) was 2 s long (100msec fade in/out) and amplitude-normalized using root mean square (RMS) method.

The final acoustic stimulus set included 24 sounds from 8 different categories (human vocalization, human non-vocalization, birds, mammals, tools, graspable objects, environmental scenes, big mechanical objects) that could be reduced to 4 superordinate categories (human, animals, manipulable objects, big objects/places) (see *Figure 1* and *Supplementary file 1*).

We created a visual version of the stimuli set. The images for the visual experiment were colored pictures collected from Internet and edited using *GIMP* (https://www.gimp.org). Images were placed on a gray (129 RGB) 400 × 400 pixels background.

## Procedure

The experimental session was divided into two parts: first the subjects underwent the fMRI experiment and then they performed a behavioral rating judgment task on the same stimuli used in the fMRI experiment.

## Similarity rating

The behavioral experiment aimed to create individual behavioral dissimilarity matrices to understand how the participants perceived the similarity of our stimulus space. Due to practical constraints, only a subset of our participants underwent the behavioral experiment (15 EBa, 11 SCa, and 9 SCv). We created each possible pair from the 24 stimuli set leading to a total of 276 pairs of stimuli. In the auditory experiment, participants heard each sound of a pair sequentially and were asked to judge from 1 to 7 how similar the two stimuli producing these sounds were. In the visual experiment, we presented each pair of stimuli on a screen to the participants and we asked them to judge from 1 to 7 how similar the two stimuli were. Since their rating was strongly based on the categorical features of the stimuli, we used the data from the behavioral experiment to build the categorical models for the representational similarity analysis (see the section '*Representational similarity analysis: correlation with representational low-level/behavioral models*').

## fMRI experiment

Each participant took part in only one experiment, either in the auditory or in the visual version. We decided to include two separate groups of sighted people, one for each modality, for two crucial reasons. First, we wanted to limit as much as possible the possibility of triggering mental imagery from one modality to the other. Second, since cross-group comparisons of representational dissimilarity analyses represent a core component of our analyses stream, we wanted to ensure a cross-group variance comparable between the blind versus the sighted and the sighted in audition versus the sighted in vision.

The procedure for the two experiments was highly similar (*Figure 1A*).

Before entering the scanner, all the stimuli (either auditory or visual) were presented to each participant to ensure perfect recognition. In the fMRI experiment each trial consisted of the same stimulus repeated twice. Rarely (8% of the occurrences), a trial was made up of two different consecutive stimuli (catch trials). Only in this case participants were asked to press a key with the right index finger if the second stimulus belonged to the living category and with their right middle finger if the second stimulus belonged to the non-living category. This procedure ensured that the participants attended and processed the stimuli. In the auditory experiment, each pair of stimuli lasted 4 s (2 s per stimulus) and the inter-stimulus interval between one pair and the next was 2 s long for a total of 6 s for each trial (*Figure 1A*). In the visual experiment, each pair of stimuli lasted 2 s (1 s per stimulus) and the inter-stimulus interval between one pair and the next was 2 s long for a total of 4 s for each trial (*Figure 1A*).

The use of a ''quick'' event-related fMRI paradigm balances the need for separable hemodynamic responses and the need for presenting many stimuli in the limited time-span of the fMRI experiment. Within both the auditory and the visual fMRI sessions, participants underwent five runs. Each run contained 3 repetitions of each of the 24 stimuli, eight catch trials and two 20s-long rest periods (one in the middle and another at the end of the run). The total duration of each run was 8 min and 40 s for the auditory experiment and 6 min for the visual experiment. For each run, the presentation of trials was pseudo-randomized: two stimuli from the same category were never presented in subsequent trials. The stimulus delivery was controlled using the Psychophysics toolbox implemented in Matlab R2012a (The MathWorks).

## fMRI data acquisition and preprocessing

We acquired our data on a 4T Bruker Biospin MedSpec equipped with an eight-channel birdcage head coil. Functional images were acquired with a T2*-weighted gradient-recalled echo-planar imaging (EPI) sequence (TR, 2000 ms; TE, 28 ms; flip angle, 73°; resolution, 3 × 3 mm³; 30 transverses slices in interleaved ascending order; 3 mm slice thickness; field of view (FoV) 192 × 192 mm²). The four initial scans were discarded for steady-state magnetization. Before each EPI run, we performed an additional scan to measure the point-spread function (PSF) of the acquired sequence, including fat saturation, which served for distortion correction that is expected with high-field imaging.

A structural T1-weighted 3D magnetization prepared rapid gradient echo sequence was also acquired for each subject (MP-RAGE; voxel size 1 × 1×1 mm³; GRAPPA acquisition with an acceleration factor of 2; TR 2700 ms; TE 4,18 ms; TI (inversion time) 1020 ms; FoV 256 mm; 176 slices).

To correct for distortions in geometry and intensity in the EPI images, we applied distortion correction on the basis of the PSF data acquired before the EPI scans (*Zeng and Constable, 2002*). Raw functional images were pre-processed and analyzed with SPM8 (Welcome Trust Centre for Neuroimaging London, UK (https://www.fil.ion.ucl.ac.uk/spm/software/spm8/) implemented in MATLAB R2013b (MathWorks). Pre-processing included slice-timing correction using the middle slice as reference, the application of temporally high-pass filtered at 128 Hz and motion correction.

## Regions of interest

Since we were interested in the brain representation of different categories we decided to focus on the ventro-occipito temporal cortex as a whole. This region is well known to contain several distinct macroscopic brain regions known to prefer a specific category of visual objects including faces, places, body parts, small artificial objects, etc. (*Kanwisher, 2010*). We decided to focus our analyses on a full mask of VOTC, and not in specific sub-parcels because we were interested in looking at the categorical representation across categories and not at the preference of a specific category compared to the others. Our study therefore builds upon the paradigm shift of viewing VOTC as a distributed categorical system rather than a sum of isolated functionally specific areas, which reframes how we should expect to understand those areas (*Haxby et al., 2001*). In fact, our main aim was to investigate how sensory input channel and visual experience impact on the general representation of different categories in the brain. Looking at one specific category-selective region at time would not allow us to address this specific question, since we would expect an imbalanced representation of the preferred category compared to the others. Indeed, to tackle our question, we need to observe the distributed representation of the categories over the entire ventral occipito-temporal cortex (*Haxby et al., 2001*). This approach has already been validated by previous studies that investigated

the categorical representation in the ventral-occipito temporal cortex, using a wide VOTC mask, such as *van den Hurk et al. (2017)*; *Kriegeskorte et al. (2008b)*; *Wang et al. (2015)*. We also added the early visual cortex (EVC) as a control node. We decided to work in a structurally and individually defined mask of VOTC using the Desikan-Killiany atlas (*Desikan et al., 2006*) implemented in FreeSurfer (http://surfer.nmr.mgh.harvard.edu). Six ROIs were selected in each hemisphere: Pericalcarine, Cuneus and Lingual areas were combined to define the early visual cortex (EVC) ROI; Fusiform, Parahippocampal and Infero-Temporal areas were combined to define the ventral occipito-temporal (VOTC) ROI. Then, we combined these areas in order to obtain one bilateral EVC ROI and one bilateral VOTC ROI (*Figure 3A*).

Our strategy to work on a limited number of relatively large brain parcels has the advantage to minimize unstable decoding results collected from small regions (*Norman et al., 2006*) and reduce multiple comparison problems intrinsic to neuroimaging studies (*Etzel et al., 2013*). All analyses, except for the topographical selectivity map (see below), were carried out in subject space for enhanced anatomico-functional precision and to avoid spatial normalization across subjects.

## General linear model

The pre-processed images for each participant were analyzed using a general linear model (GLM). For each of the five runs, we included 32 regressors: 24 regressors of interest (each stimulus), 1 regressor of no-interest for the target stimulus, six head-motion regressors of no-interest and one constant. From the GLM analysis, we obtained a β-image for each stimulus (i.e. 24 sounds) in each run, for a total of 120 (24 stimuli x five runs) beta maps.

## Topographical selectivity map

For this analysis, we needed all participants to be coregistered and normalized in a common volumetric space. To achieve maximal accuracy, we relied on the DARTEL (Diffeomorphic Anatomical Registration Through Exponentiated Lie Algebra; *Ashburner, 2007*) toolbox. DARTEL normalization takes the gray and white matter templates from each subject to create an averaged template based on our own sample that will be used for the normalization. The creation of a study-specific template using DARTEL was performed to reduce deformation errors that are more likely to arise when registering single subject images to an unusually shaped template (*Ashburner, 2007*). This is particularly relevant when comparing blind and sighted subjects given that blindness is associated with significant changes in the structure of the brain itself, particularly within the occipital cortex (*Dormal et al., 2016*; *Jiang et al., 2009*; *Pan et al., 2007*; *Park et al., 2009*).

To create the topographical selectivity map (*Figure 1B*), we extracted in each participant the β-value for each of our four main conditions (animals, humans, manipulable objects and places) from each voxel inside the VOTC mask and we assigned to each voxel the condition producing the highest β-value (winner takes all). This analysis resulted in specific clusters of voxels that spatially distinguish themselves from their surround in terms of selectivity for a particular condition (*van den Hurk et al., 2017*; *Striem-Amit et al., 2018a*).

Finally, to compare how similar are the topographical selectivity maps in the three groups we followed, for each pair of groups (i.e. 1.SCv-EBa; 2.SCv-SCa; 3.EBa-SCa) these steps: (1) We computed the Spearman's correlation between the topographical selectivity map of each subject from Group one with the averaged selectivity map of Group two and we compute the mean of these values. (2) We computed the Spearman's correlation between the topographical selectivity map of each subject from Group two with the averaged selectivity map of Group one and we computed the mean of these values. (3) We averaged the two mean values obtained from step 1 and step 2, in order to have one mean value for each group comparison (see the section 'Statistical analyses' for details about the assessment of statistical differences).

We ran this analysis using the four (*Figure 1B*) and the eight categories (see *Figure 1—figure supplement 1*) and both analyses lead to almost identical results. We decided to represent the data of the four main categories for simpler visualization of the main effect (topographical overlap across modalities and groups).

In order to go beyond the magnitude of the correlation between the topographical selectivity maps and to explore the quality of the (dis)similarity between the topographical maps of our subjects and groups we computed the Jaccard index between them. The Jaccard similarity coefficient is

a statistic used for measuring the similarity and diversity of sample sets (*Devereux et al., 2013*; *Xu et al., 2018*). The Jaccard coefficient is defined as the size of the intersection divided by the size of the union of the sample sets. This value is 0 when the two sets are disjoint, 1 when they are equal, and between 0 and 1 otherwise.

First, we looked at the consistency of the topographical representation of the categories across subjects within the same group. For this within group analysis we computed the Jaccard similarity between the topographical selectivity map of each subject and the mean topographical map of his own group. This analysis produces 4 Jaccard similarity indices (one for each of the main category: (1) animals, (2) humans, (3) manipulable objects and (4) big objects and places) for each group (see *Figure 2B*). Low values for one category within a group mean that the topographical representation of that category varies a lot across subjects of that group. Statistical significance of the Jaccard similarity index within groups was assessed using parametric statistics: One sample t-tests to assess the difference against zero and a repeated measure ANOVA (4 categories * 3 groups) to compare the different categories and groups.

In addition, we computed the Jaccard similarity index for each category, between the topographical map of each blind and sighted subject in the auditory experiment (EBa and SCa) with the averaged topographical selectivity map from the sighted in the visual experiment (SCv; see *Figure 2C* for the results). In more practical terms, this means that we have 4 Jaccard similarity indices (one for each of the main category: (1) animal, (2) human, (3) manipulable objects and (4) big objects and places) for each of the two groups' pairs: EBa-SCv and SCa-SCv. These values can help in explore more in detail the similarity and differences among the topographical representations of our categories when they are presented visually compared to when they are presented acoustically, both in sighted and in blind. Statistical significance of the Jaccard similarity index between groups was assessed using parametric statistics: one sample T-tests to test the difference against zero, a repeated measure ANOVA (4 categories * 2 groups) to compare the different categories and groups. The Greenhouse-Geisser sphericity correction was applied.

However, the Jaccard similarity values could be partially driven by the number of voxels that show selectivity for each category. For instance, the absence of overlap between the voxels that prefer animals in EBa and SCv could be explained by the fact that different voxels prefer animals in the two groups or by the fact that one or both groups have a limited number of voxels that show a preference for this category. To disentangle these two possibilities we counted in each group the number of voxels within VOTC that prefers each category (see *Figure 2A* for the results). We ended up with four values (one number for each category) for each group. Statistical significance of the number of voxels within groups was assessed using parametric statistics: One sample t-tests to assess the difference against zero, a repeated measure ANOVA (four categories * three groups) to compare the different categories and groups.

## MVP-classifications: Binary decoding

We performed a binary MVP-classification (using SVM - support vector machine classifier) to look at the ability of each ROI to distinguish between two categories at time. With eight categories we can have 28 possible pairs, resulting in 28 binary MVP-classification tests in each ROI. Statistical significance of the binary classification was assessed using t-test against the chance level. We, then, averaged the 28 accuracy values of each subject in order to have one mean accuracy value for subject. Statistical significance of the averaged binary classification was assessed using parametric statistics: t-test against zero and ANOVA.

## Representational similarity analysis (RSA): Correlation between neural dissimilarity matrices of the three groups

We further investigated the functional profile of the ROIs using RSA. RSA was performed using the CoSMoMVPA toolbox, implemented in Matlab (r2013b; Matworks). The basic concept of RSA is the dissimilarity matrix (DSM). A DSM is a square matrix where the number of columns and rows corresponds to the number of the conditions ($8 \times 8$ in this experiment) and it is symmetrical about a diagonal of zeros. Each cell contains the dissimilarity index between the two stimuli. We used the binary MVP-classification as dissimilarity index to build neural DSMs (*Carlson et al., 2013*; *Cichy and Pantazis, 2017*; *Cichy et al., 2013*; *Dobs et al., 2019*; *Haxby et al., 2014*; *Haxby et al., 2011*;

*O'Toole et al., 2005*; *Pereira et al., 2009*; *Proklova et al., 2019*) for each group, in order to compare the functional profile of the ROIs between the three groups. In this way, we ended up with a DSM for each group for every ROI.

We preferred to use binary MVP-classification as dissimilarity index to build neural DSMs rather than other types of dissimilarity measures (e.g. Pearson correlation, Euclidean distance, Spearman correlation) since two experimental conditions that do not drive a response and therefore have uncorrelated patterns (noise only, $r \approx 0$) appear very dissimilar ($1 – r \approx 1$). When using a decoding approach instead, due to the intrinsic cross-validation steps, we would find that the two conditions that don't drive responses are indistinguishable, despite their substantial correlation distance (*Walther et al., 2016*) since the noise is independent between the training and testing partitions, therefore cross-validated estimates of the distance do not grow with increasing noise. This was crucial in our study since we are looking at brain activity elicited by sounds in brain regions that are primarily visual (EVC and VOTC), therefore the level of noise is expected to be high, at least in sighted people.

Finally, to compare how similar are the DSMs in the three groups, for each pair of groups (i.e. 1. SCv-EBa; 2.SCv-SCa; 3.EBa-SCa) (1) we computed the Spearman's correlation between the upper triangular DSM (excluding the diagonal) of each subject from Group one with the averaged upper triangular DSM (excluding the diagonal) of Group two and we compute the mean of these values. (2) We computed the Spearman's correlation between the upper triangular DSM (excluding the diagonal) of each subject from Group two with the averaged upper triangular DSM (excluding the diagonal) of Group one and we computed the mean of these values. (3) We averaged the two mean values obtained from step 1 and step 2, in order to have one mean value for each groups' comparison.

Considering the unidirectional hypothesis for this test (a positive correlation between neural similarity and models similarity) and the difficult interpretation of a negative correlation, one-tailed statistical tests were used. For all other tests (e.g., differences between groups), for which both directions might be hypothesized, two-tailed tests were applied (*Peelen et al., 2014*; *Evans et al., 2019*).

See the section '*Statistical analyses*' for details about the assessment of statistical differences.

## Hierarchical clustering analysis on the brain categorical representations

In order to go beyond the correlation values and to explore more qualitatively the representational structure of VOTC and EVC in the three groups, we implemented a hierarchical clustering approach (*King et al., 2019*). First, we created a hierarchical cluster tree for each brain DSM using the 'linkage' function in Matlab, then we defined clusters from this hierarchical cluster tree with the 'cluster' function in Matlab. Hierarchical clustering starts by treating each observation as a separate cluster. Then, it identifies the two clusters that are closest together, and merges these two most similar clusters. This continues until all the clusters are merged together or until the clustering is 'stopped' to a n number of clusters. We repeated the clustering three times for each DSM, stopping it at 2, 3 and 4 clusters, respectively (see *Figure 5*). In this way, we could compare the similarities and the differences of the clusters at the different scales across the groups. We applied the same clustering analysis also on the behavioral data (see *Figure 5—figure supplement 1*).

## Representational similarity analysis (RSA): correlation with representational low-level/behavioral models

We then intended to investigate which features of the visual and auditory stimuli were represented in the different ROIs of sighted and blind subjects. RSA allows the comparisons between the brain DSMs extracted from specific ROIs with representational DSMs, based on physical properties of the stimuli or based on behavioral rating of the perceived categorical similarity of our stimuli.

## Low-level DSM in the auditory experiment: pitch DSM

Pitch corresponds to the perceived frequency content of a stimulus. We selected this specific low-level auditory feature for two reasons. First, previous studies showed that this physical property of the sounds is distinctly represented in the auditory cortex and may create some low-level bias of auditory category selective responses in the temporal cortex (*Giordano et al., 2013*; *Leaver and Rauschecker, 2010*; *Moerel et al., 2012*). Second, we confirmed with our own SCa group that,

among alternative auditory RDMs based on separate acoustic features (e.g. Harmonicity on noise ratio, Spectral centroid), the pitch model correlated most with brain RDM extracted from the temporal cortex. This provided strong support that this model was maximally efficient in capturing the encoding of sounds based on acoustic features in auditory cortical regions (see *Figure 6—figure supplement 1*).

We computed a pitch value for each of the 24 auditory stimuli, using the Praat software and an autocorrelation method. This method extracts the strongest periodic component of several time windows across the stimulus and averages them to have one mean pitch value for that stimulus. The 'pitch floor' determines the size of the time windows over which these values are calculated in Praat. Based on a previous study, we chose a default pitch floor of 60 Hz (*Leaver and Rauschecker, 2010*). We then averaged the pitch values across stimuli belonging to the same category. Once we obtained one pitch value for each category, we built the DSM computing the absolute value of the pitch difference for each possible pairwise (see *Figure 6A*). The pitch DSM was not positively correlated with the behavioral DSM of neither SCa (r=–0.36, p=0.06) nor EBa (r = –0.29, p=*0.13*).

## Low-level DSM in the visual experiment: Hmax- C1 model

The Hmax model (*Serre et al., 2007*) reflects the hierarchical organization of the visual cortex (*Hubel and Wiesel, 1962*) in a series of layers from V1 to infero-temporal (IT) cortex. To build our low-level visual model we used the output from the V1- complex cells layer. The inputs for the model are the gray-value luminance images presented in the sighted group doing the visual experiment. Each image is first analysed by an array of simple cells (S1) units at 4 different orientations and 16 scales. At the next C1 layer, the image is subsampled through a local Max pooling operation over a neighbourhood of S1 units in both space and scale, but with the same preferred orientation (*Serre et al., 2007*). C1 layer stage corresponds to V1 cortical complex cells, which shows some tolerance to shift and size (*Serre et al., 2007*). The outputs of all complex cells were concatenated into a vector as the V1 representational pattern of each image (*Khaligh-Razavi and Kriegeskorte, 2014*; *Kriegeskorte et al., 2008a*). We averaged the vectors of images from the same category in order to have one vector for each category. We, finally, built the (8 × 8) DSM computing 1- Pearson's correlation of each pair of vectors (see *Figure 6A*). The Hmax-C1 DSM was significantly correlated with the SCv behavioral DSM (r = 0.56, p=*0.002*).

## Behavioral-categorical DSMs

We used the pairwise similarity judgments from the behavioral experiment to build the semantic DSMs. We computed one matrix for each subject that took part in the behavioral experiment and we averaged all the matrices of the participants from the same group to finally obtain three mean behavioral-categorical DSMs, one for each group (i.e. EBa, SCa, SCv; *Figure 4A*). The three behavioral matrices were highly correlated between them (SCv-EBa: r = 0.89, p<0.001; SCv-SCa: r = 0.94, p<0.001; EBa-SCa: r = 0.85, p<0.001), and the similarity judgment was clearly performed on a categorical-membership basis (*Figure 6A*).

The last step consisted in comparing neural and external DSMs models using a second order correlation. Because we wanted to investigate each representational model independently from the other, we relied on Spearman's rank partial correlation: in the auditory experiment, we removed the influence of the pitch similarity when we were computing the correlation with the behavioral matrix, and vice versa; in the visual experiment, we removed the influence of the Hmax-C1 model similarity, when we were computing the correlation with the behavioral matrix, and vice versa. In this way, we could measure the partial correlation for each external model for each participant separately. Importantly, we did not correlate the full symmetrical DSMs but only the upper triangular DSM excluding the diagonal.

See the section '*Statistical analyses*' for details about the assessment of statistical differences.

## RSA: Inter-subjects correlation

To examine the commonalities of the neural representational space across subjects in VOTC, we extracted the neural DSM of every subject individually and then correlated it with the neural DSM of every other subject. Since we have 49 participants in total, this analysis resulted in a 49 × 49 matrix (*Figure 7*) in which each line and column represents the correlation of one subject's DSM with all

other subjects' DSM. The three main squares in the diagonal (*Figure 7*) represent the within group correlation of the 3 groups. We averaged the value within each main square (only the upper half excluding the diagonal) on the diagonal to obtain a mean value of within group correlation for each group. The three main off diagonal squares (*Figure 7*) represent the between groups correlation of the three possible groups' pairs (i.e. 1.SCv/EBa; 2.SCv-SCa; 3.EBa-SCa). We averaged the value within each main off-diagonal square in order to obtain a mean value of between groups correlation for each groups' pair.

See the section '*Statistical analyses*' for details about the assessment of statistical differences.

## Representational connectivity analysis

Representational connectivity analysis were implemented to identify the representational relationship among the ROIs composing VOTC and the rest of the brain (*Kriegeskorte et al., 2008a*; *Pillet et al., 2018*). This approach can be considered a type of connectivity where similar RDMs of two ROIs indicate shared representational structure and therefore is supposed to be a proxy for information exchange (*Kriegeskorte et al., 2008b*). Representational connectivity between two ROIs does not imply a direct structural connection but can provide connectivity information from a functional perspective, assessing to what extent two regions represent information similarly (*Xue et al., 2013*).

To perform this analysis, we included 30 bilateral parcels (covering almost the entire cortex) extracted from the segmentation of individual anatomical scan following the Desikan-Killiany atlas implemented in FreeSurfer (http://surfer.nmr.mgh.harvard.edu). We only excluded three parcels (Entorhinal cortex, Temporal Pole and Frontal Pole) because their size was too small and signal too noisy (these regions are notably highly susceptible to signal drop in EPI acquisition) to allowed the extraction of reliable dissimilarity matrices in most of the participants. We merged together the left and right corresponding parcels in order to have a total of 30 bilateral ROIs for each subject. From each ROI we extracted the dissimilarity matrix based on binary decoding accuracies, as described in the section '*Representational similarity analysis (RSA): Correlation between neural dissimilarity matrices of the three groups*'. Finally, we computed the Spearman's correlation between the three seed ROIs (i.e. fusiform gyrus, parahippocampal gyrus and infero-temporal cortex) and all the other 27 ROIs. We ended up with a connectivity profile of 3 (number of seeds) by 27 (ROIs representing the rest of the brain) for each subject. We considered this 3*27 matrix as one representational connectivity profile of the seed region in each subject.

Finally, to compare how similar are the RSA connectivity profiles in the three groups, for each pair of groups (i.e. 1.SCv-EBa; 2.SCv-SCa; 3.EBa-SCa): (1) We computed the Spearman's correlation between the representational connectivity profile of each subject from Group one with the averaged representational connectivity profile from Group two and we compute the mean of these values. (2) We computed the Spearman's correlation between the representational connectivity profile of each subject from Group two with the averaged representational connectivity profile of Group one and we computed the mean of these values. (3) We averaged the two mean values obtained from step 1 and step 2, in order to have one mean value for each group comparison.

We computed the same analysis also in the EVC, as a control ROI. In this case the three nodes ROIs were the pericalcarine cortex, the cuneus and the lingual gyrus.

See the section '*Statistical analyses*' for details about the assessment of statistical differences.

## Statistical analyses

To assess statistical differences, we applied parametric tests (T-Test and ANOVA) in the analyses that met the main assumptions required by parametric statistics: normal distribution of the data and independency of the observations. Moreover, in case of statistical comparisons between different groups we ran the Levene's test to check for the assumption of equality of variances between the groups, in case the test was significant (suggesting different levels of variance) we applied the Welch Homogeneity correction. Parametric tests were used in the Jaccard similarity analyses (both within and between groups), in the analysis on the selective voxel's count and in the averaged binary decoding analysis. In all these analyses the correlation data were z-transformed before subjecting them to parametric statistics.

However, the correlation of the topographical selectivity maps, the correlation of the brain dissimilarity matrices, the correlation of the RSA connectivity profiles and the inter-subject DSMs correlation did not meet the assumption of independency of the data. In fact, in these analyses we contrast group's comparisons, so data from the same subjects are always included in two comparisons (e.g. data from EBa subjects are included both in the SCv-EBa and in EBa-SCa comparisons). For this reason, the use of permutation was a preferable approach in the case of these analyses. In each of this analysis we have one vector of values for each subject in each ROI (i.e. The vector containing the categorical selectivity label of each voxel; The brain dissimilarity matrix in the format of pairwise distance vector; The vectorized RSA connectivity profile). In each analysis we want to correlate these values between each possible pair of groups, which are three in total: SCv-EBa; SCv-SCa; EBa-SCa. To compute the average correlation value between each pair of groups we followed these steps: (1) We computed the Spearman's correlation between the vector of each subject from Group one with the mean vector of Group two and we computed the mean of these values (e.g. we correlated the vector from each EBa subject with the mean vector from the SCv group). (2) We computed the Spearman's correlation between the vector of each subject from Group two with the mean vector of Group one and we computed the mean of these values (e.g. we correlated the vector from each SCv sub. with the mean vector from the EBa group). (3) We averaged the two mean values obtained from step 1 and step 2, in order to have one mean value for each group comparison. Since our data points are not completely independent, we cannot use parametric statistics (*Parsons et al., 2018*). Therefore, to test statistical differences we used a permutation test (10.000 iterations): (4) We randomly permuted the conditions of the vector of each subject from Group 1 and of the mean vector of Group 2 and we computed the correlation (as in Step 1). (5) We randomly permuted the conditions of the vector of each subject from Group 2 and of the mean vector of Group 1 and we computed the correlation (as in Step 2). (6) We averaged the 2 mean values obtained from step 4 and step 5. (7) We repeated these steps 10.000 times to obtain a distribution of correlations simulating the null hypothesis that the two vectors are unrelated (*Kriegeskorte et al., 2008b*). If the actual correlation falls within the top $\alpha \times 100\%$ of the simulated null distribution of correlations, the null hypothesis of unrelated vectors can be rejected with a false-positives rate of $\alpha$.

Only in the case of the correlation of topographical maps, we constrained the permutation performed in the step five in order to take into consideration the inherent smoothness/spatial dependencies in the univariate fMRI data. In each subject, we individuated each cluster of voxels showing selectivity for the same category and we kept these clusters fixed in the permutation, assigning randomly a condition to each of these predefined clusters. In this way, the spatial structure of the topographical maps was kept identical to the original one, making very unlikely that a significant result could be explained by the voxels' spatial dependencies. We may however note that this null-distribution is likely overly conservative since it assumes that size and position of clusters could be created only from task-independent spatial dependencies (either intrinsic to the acquisition or due to smoothing). We had to exclude one EBa subject from the analysis because he had less than seven clusters in his topographical map, which is not enough to have 10000 combinations needed for the permutation given our four categories tested (possible combinations = n_categories$^{n\_clusters}$; $4^7 = 16384$).

To test the difference between the group pairs' correlations (e.g. to test if the correlation between SCv and EBa was different from the correlation of SCv and SCa), we used a permutation test (10.000 iterations): (8) We computed the difference between the correlation of Pair one and Pair 2: mean correlation Pair1 – mean correlation Pair2. (9) We kept fixed the labels of the group common to the two pairs and we shuffled the labels of the subjects from the other two groups (e.g. if we are comparing SCv-EBa versus SCv-SCa, we keep the SCv group fixed and we shuffle the labels of EBa and SCa). (10) After shuffling the groups' labels we computed again the point 1-2-3 and 8. (11) We repeated this step 10.000 times to obtain a distribution of differences simulating the null hypothesis that there is no difference between the two pairs' correlations. If the actual difference falls within the top $\alpha \times 100\%$ of the simulated null distribution of difference, the null hypothesis of absence of difference can be rejected with a false-positives rate of $\alpha$.

Finally, also the RSA with representational low-levels and behavioral models did not meet the assumption of independency of the data. In fact, for dissimilarity matrices the independence of the samples cannot be assumed, because each similarity is dependent on two response patterns, each of which also codetermines the similarities of all its other pairings in the RDM (*Kriegeskorte et al.,*

*2008b*). For each group, the statistical difference from zero was determined using permutation test (10000 iterations), building a null distribution for these correlation values by computing them after randomly shuffling the labels of the matrices. Similarly, the statistical difference between groups was assessed using permutation test (10000 iterations) building a null distribution for these correlation values by computing them after randomly shuffling the group labels. Moreover, considering the uni-directional hypothesis for this test (a positive correlation between neural similarity and models similarity) and the difficult interpretation of a negative correlation, one-tailed statistical tests were used. For all other tests (e.g., differences between groups), for which both directions might be hypothesized, two-tailed tests were used (*Peelen et al., 2014*; *Evans et al., 2019*).

In each analysis, all the p-values are reported after false discovery rate (FDR) correction implemented using the matlab function 'mafdr'.

## Acknowledgements

We would like to express our gratitude to Marco Barilari, Stefania Benetti, Giorgia Bertonati, Francesca Barbero who have helped with the data acquisition, to Yangwen Xu, Matthew Bennet and Remi Gau for giving comments on a preliminary version of the paper, to Jorge Jovicich for helping to set-up the fMRI acquisition parameters and to Pietro Chiesa for continuing support with stimuli-delivery systems. We are also extremely thankful to our blind participants and to the Unioni Ciechi of Trento, Mantova, Genova, Savona, Cuneo, Torino, Trieste and Milano and the blind Institute of Milano for helping with the recruitment. The project was funded by the ERC starting grant MADVIS (Project: 337573) and the Belgian Excellence of Science program (Project: 30991544) awarded to Olivier Collignon. Olivier Collignon is research associate at the Fond National de la Recherche Scientifique de Belgique (FRS-FNRS).

## Additional information

### Funding

| Funder | Grant reference number | Author |
| --- | --- | --- |
| European Commission | Starting Grant MADVIS: 337573 | Olivier Collignon |
| Excellence of Science | 30991544 | Olivier Collignon |
| Fonds De La Recherche Scientifique - FNRS | | Olivier Collignon |

The funders had no role in study design, data collection and interpretation, or the decision to submit the work for publication.

### Author contributions

Stefania Mattioni, Conceptualization, Data curation, Software, Formal analysis, Investigation, Visualization, Methodology; Mohamed Rezk, Ceren Battal, Methodology; Roberto Bottini, Conceptualization; Karen E Cuculiza Mendoza, Software, Methodology; Nikolaas N Oosterhof, Data curation, Software, Methodology; Olivier Collignon, Conceptualization, Resources, Data curation, Software, Supervision, Funding acquisition, Validation, Visualization, Methodology, Project administration

### Author ORCIDs

Stefania Mattioni (iD) https://orcid.org/0000-0001-8279-6118
Mohamed Rezk (iD) https://orcid.org/0000-0002-1866-8645
Ceren Battal (iD) https://orcid.org/0000-0002-9844-7630
Roberto Bottini (iD) http://orcid.org/0000-0001-7941-7762
Olivier Collignon (iD) https://orcid.org/0000-0003-1882-3550

### Ethics

Human subjects: The ethical committee of the University of Trento approved this study (protocol 2014-007) and participants gave their informed consent before participation.

Decision letter and Author response

Decision letter https://doi.org/10.7554/eLife.50732.sa1

Author response https://doi.org/10.7554/eLife.50732.sa2

---

## Additional files

### Supplementary files

- Supplementary file 1. Categories and stimuli description.
- Supplementary file 2. Characteristics of early blind participants.
- Transparent reporting form

### Data availability

Processed data have been made available on OSF at the link https://osf.io/erdxz/. To preserve participant anonymity and due to restrictions on data sharing in our ethical approval, fully anonymised raw data can only be shared upon request to the corresponding author.

The following dataset was generated:

| Author(s) | Year | Dataset title | Dataset URL | Database and Identifier |
|---|---|---|---|---|
| Mattioni S | 2020 | Categorical representation from sound and sight in the ventral occipito-temporal cortex of sighted and blind - open data | https://osf.io/erdxz/ | Open Science Framework, erdxz |

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
