## [Decision Letter]

**Acceptance summary:**

The study adds important support, using brain decoding, to the theory that the visual cortex in early blindness processes similar category boundaries as in vision. This is an extremely well designed study and extends beyond previous research in the field by adopting a substantially richer stimulus space, and by carefully considering alternative contributions to the stimulus domains. Together with detailed and thoughtful analysis, the study provides a comprehensive account of the similarities and differences of categorical representation in the blind ventral occipito-temporal cortex.

**Decision letter after peer review:**

Thank you for submitting your article "Similar categorical representation from sound and sight in the occipito-temporal cortex of sighted and blind" for consideration by *eLife*. Your article has been reviewed by three peer reviewers, including Tamar R Makin as the Reviewing Editor and Reviewer #1, and the evaluation has been overseen by Barbara Shinn-Cunningham as the Senior Editor. The reviewers have opted to remain anonymous.

The reviewers have discussed the reviews with one another and the Reviewing Editor has drafted this decision to help you prepare a revised submission.

Summary

This study is aimed at assessing categorical representational similarities between sighted and blind individuals in the early and ventral occipitotemporal visual cortex. The authors used sounds of objects, animals, humans and scenes to construct a representational structure in the blinds visual cortex and compared that to the representational structure evoked by images and sounds to the same categories in sighted individuals. They showed that the blind response pattern and response pattern connectivity for such sounds is broadly similar to that found in sighted using images in the VOTC, and that this representational structure didn't merely reflect auditory pitch similarity or low-level visual processing. Additionally, they showed that the inter-subject consistency is higher in the blind for sounds than in the sighted for the same sounds, but lower than for sight. Overall the study adds important support, using RSA, to the theory that the visual cortex in early blindness processes similar category boundaries as in vision. This is an extremely well designed study and extends beyond previous research in the field by adopting a richer stimulus space and more detailed analyses that provide a more comprehensive account of the similarities and differences between the different groups and stimulus domains.

All three reviewers were particularly enthusiastic about the experimental design and were excited towards the opportunity of developing a more comprehensive understanding of the similarities and differences between blind and sighted individuals. In that respect, we felt that although interesting, the results highlighting the similarities across groups shouldn't obstruct the authors from exploring potential differences between groups. In addition, multiple questions were raised relating to the specific statistical tests and ROI definition, which required further justification or better clarity of the test specifics. In some cases, the manuscript would benefit from more mainstream/standardised analyses, as there were concerns that the current approaches were at times inflating the effects/their significance. The consensus across reviewers was that we are interested in a more nuanced understanding of your findings, even if this might potentially weaken some of your key conclusions (i.e. similar categorical representation from sound and sight in the occipito-temporal cortex of sighted and blind).

Major comments

1) The reviewers felt that there are some missed opportunities here to explore beyond the simple DSM correlations. One reviewer suggested more formal model comparisons, another suggested interrogating the representation structure with more detail. I'm appending the specific comments below:

Reviewer 1: In Abstract and elsewhere (e.g. “the representational structure of visual and auditory categories was almost double in the bind when compared to the sighted group") – The authors are confusing the R with the effect size (R square) which is actually almost four times greater. In general – they might like to consider more formal comparison across models (PCM or equivalent), while accounting for noise levels (if the inter-subject variance is greater than the correlation is likely to be lower).

Reviewer 3: The authors suggest that "VOTC reliably encodes sounds categories in blind people using a representational structure strikingly similar to the one found in vision". This conclusion is based on the magnitude of correlations between DSMs (e.g. Figure 3). However, I think it is important to go beyond the significant correlations and look carefully at the representational structure (e.g. see King et al., 2019). For example, while the dendrogram for SCv shows an initial split between the animate and inanimate items, for EBa the initial split is for the human conditions (HV, HN) from all others, and the animal conditions (AM, AM) and more closely grouped with BM and MG. Thus, the high correlation between SCv and EBa actually belies some differences in the representational structure that I think are worth commenting on. Similar comparison can be made for EVC and for SCa. So, I recommend the authors consider the representational structure in more detail and probably should draw back slightly from the claim that the representational structure in EBa and SCv are "strikingly similar".

2) One of the reviewers was particularly concerned about the usage of group means to characterise differences between groups, without adequately representing inter-subject variance within each group. In some analyses (e.g. the topographical selectivity map) the mean is taken as the sole group statistics (as a fixed effect), making the analysis highly susceptible to outliers and precluding generalisation to other samples. In other analyses (e.g. DSMs) a permutation test is included to determine group differences; however the specifics of this permutation tests are not well explained and require more thought – does this permutation test adequately captures the variance of the categorical representation between sighted and blind individuals? Or are the authors potentially confusing their units of analysis (see Parsons et al., *eLife*, 2018)? Ideally, variance should be assessed by measuring the effect in each individual participant and comparing the effect across groups (indeed this approach is used in some of the tests in the current study). In the case of DSM, each participant (e.g. from the EBa group) should be correlated to the SCv group, generating a distribution of the effect of interest. To an extent, a similar analysis is presented in Figure 5, providing much lower effect sizes than highlighted throughout the paper. So, we are looking for a more considered quantification of the main effects. A lot of confusion/misinterpretation could be avoided if the authors used standard statistics that takes into consideration the inter-subject variance, which is the key unit of your analysis (e.g. t-test and its variants). If the authors feel that their own approach is preferable, then they should increase the clarity on their analysis and its validity, with a clear statement of what null is.

3) A few of the reviewers were commenting on inconsistent statistical criteria, with the significance test flipping between one-tail to two-tails. For example, correlations between DSM were tested for significance using a one-tailed permutation test. This resulted in identifying a very strong negative correlation between the SCv and SCa in the early visual cortex (r = -.50), as non-significant, where in fact this effect is stronger than the correlation presented between SCv and EBa DSMs – r=.41, p=0.03). Similarly, the SCv DSM in the early visual cortex may be significantly negatively correlated to the behavioral similarity ratings (r = -0.09; the in sighted EVC DSM is reported to be significantly correlated with both the behavioural DSMs at r=.09, p=0.01). This could be an interesting finding, e.g. if cross-modal inhibition is categorically selective (allowing for decoding of sound categories in EVC in another study; (Vetter et al., 2014)). This finding could also add to the mechanistic explanation of the plasticity in the blind. But at present it is uninterpretable. We therefore request that one-tail testing would be avoided, unless there's strong statistical justification to usage it (e.g. in case of a replication of previous findings). In that case, the reasons for the one-tail hypothesis should be reported more transparently and interpreted more cautiously. But if the authors are confirming a new hypothesis, which we believe is the case for most of the tests reported, we'd encourage them to stick to two-tailed hypothesis testing. The authors are welcomed to consider correction for multiple comparisons, but again, we are looking for consistency throughout the manuscript and justification of the criteria used (or abandoned).

4) The focus on VOTC, and the specific definition of the ROIs requires further consideration. In particular the reviewers mentioned the lack of discrimination within it, limited coverage of the ventral stream, and lack of consideration of other areas which might be involved in categorical representation, e.g. high order auditory areas or other visual areas in the lateral occipital cortex (see also comment 6 below). Further exploratory analysis, focused by previous research on similar topics, would strengthen the interoperability of the results. A searchlight approach might be particularly helpful, though we leave it to the authors to decide on the specific methodology.

5) The topographic selectivity analysis raised multiple comments from the reviewers. Beyond the issue raised in comment 2 above, it was agreed that this analysis potentially reveals important differences between the groups that are not adequately captured in the current presentation and analysis. For example, Reviewer 2 mentions: "the blind show nearly no preference for animal sounds in accordance with claims made by (Bi et al., 2016); in the sighted the human non-vocalizations are the preferred category for face selective areas in the visual cortex but in the auditory version (in both blind and sighted) there is mostly vocalization selectivity; The blind show little or no preference for the large-environmental sounds whereas the sighted show a differentiation between the two sound types". Reviewer 3 mentions: "In SCv, the medial ventral temporal cortex is primarily BIG-ENV, but in EBa, that same region is primarily BIG-MEC. Similarly, there appears to be more cortex showing VOC. In SCa than in either of the other two groups". We appreciate that some of the analysis might have been used as a replication of the Van den Hurk study, but we also expect them to stand alone, in terms of quality of analysis and interpretation. In particular, these topographical preference differences could be interesting and meaningful with relation to understanding blind visual cortex function, but it is hard to judge their strength or significance as the topographical selectivity maps are not thresholded, by activation strength or selectivity significance. In addition, we would like some further clarification of how the correlations have been constructed, if indeed a use-takes-all approach is used to label each voxel.

6) In the representational connectivity analysis, the authors may be overestimating the connectivity between regions due to contribution of intrinsic fluctuations between areas. To more accurately estimate the representational connectivity, it would be better if the authors used separate trials for the seed and target regions. See Henriksson et al., 2015, Neuroimage for a demonstration and discussion of this issue. Indeed, considering the clear differences found elsewhere between groups, the high between-group correlations are striking. It could have been informative to examine a control node (e.g. low-level visual cortex; low level auditory cortex) just to gain a better sense for what these correlations reveal. Also, in this section it is not clear why the authors use a separate ROI for each of the 3 VOTC sub-regions, rather than the combination of all three, as they have for the main analysis? As a minor point, the reviewer wasn't clear how the authors switched from 3 nodes for each group to only one comparison for each group pair.

---

## [Author Response]

Major comments1) The reviewers felt that there are some missed opportunities here to explore beyond the simple DSM correlations. One reviewer suggested more formal model comparisons, another suggested interrogating the representation structure with more detail. I'm appending the specific comments below:Reviewer 1: In Abstract and elsewhere (e.g. 'the representational structure of visual and auditory categories was almost double in the bind when compared to the sighted group") – The authors are confusing the R with the effect size (R square) which is actually almost four times greater.

We thank the reviewers for highlighting this confusing statement. We agree that such claims were unclear and potentially leading to misunderstanding the outcome of correlation values and effect size. In order to avoid such confusion, we removed the sentence from the Abstract and we rephrased the sentence in the Discussion. The sentence now reads:

“The representational structure of visual categories in sighted was significantly closer to the structure of the auditory categories in blind than in sighted”.

In general – they might like to consider more formal comparison across models (PCM or equivalent), while accounting for noise levels (if the inter-subject variance is greater than the correlation is likely to be lower).

We thank the reviewers for this suggestion. We fully agree that the intersubject variance within a brain region and the consequent noise level within each group are important points to take into consideration when looking at the correlation with our models. The Pattern Component Modeling (PCM) is indeed a useful analysis to perform a formal comparison between models notably normalizing the results according to the noise ceiling (Diedrichsen et al., 2018).

However, we believe that the use of normalization according to the noise ceiling might not be the best approach in our study. A high noise ceiling is indicative of high inter-subject reliability in the representational structure of a region for our stimuli space. Therefore, if we normalize our data according to the noise ceiling, we would lose important information about the inter-subject variability itself and this could potentially bring to some erroneous conclusion. One illustrative example of a possible misinterpretation of the data after such normalization comes from the comparison of occipital and temporal brain regions in the sighted in the auditory experiment (SCa). As expected, in Author response image 1 we see much higher decoding of auditory categories in the temporal cortex (~75% decoding accuracy using mean binary decoding of our 8 categories) when compared to VOTC (~55% decoding accuracy using mean binary decoding of our 8 categories).

**Author response image 1. respfig1:** MVP-classification results in the SCa group. The 28 binary classification accuracies averaged are represented for the VOTC (in pink) and the temporal ROI (in green). Each circle represents one subject, the coloured horizontal line represents the group mean and the black vertical bars represent the standard error of the mean across subjects.

We can now go a step further and look at the RSA analysis in the TEMP and VOTC parcels. In Author response image 2, left side, you see a dot-plot graph representing the correlation of the behavioral model with the DSMs extracted from both VOTC and TEMP parcels in SCa. We see, again as expected, that the categorical model correlates more with the temporal DSM than with the VOTC DSM. In addition, the inter-subject correlation -therefore the noise ceiling- (represented by the grey line) is higher in the temporal region than in VOTC. This is an expected result, suggesting that the inter-subject variance in VOTC is higher (i.e. the noise ceiling is lower) compared to the one in the temporal region for the obvious reason that temporal regions likely have a more coherent response code across subjects given their prominent role in coding auditory information (including low-level features of the sounds that can be segregated from our different categories). Such effect has been extensively described by the work of Hasson et al., (Hasson et al., 2004, 2008), in which the authors reported a striking level of voxelby-voxel synchronization between individuals, especially in the sensory cortices involved in the processing of the stimuli at play (e.g. primary and secondary visual and auditory cortices during the presentation of visual and auditory video clips, respectively).

**Author response image 2. respfig2:** Dot-plot graphs representing the correlation of the behavioral model with the DSMs extracted from both VOTC (pink) and a TEMP (green) parcels, in sighted for auditory stimuli (SCa). Each dot represents one subject, the colored (pink and green) lines represent the group mean values. The vertical black bars represent the standard error of the mean across subjects. Left panel: Spearman’s correlation with the behavioural model is represented here. The grey lines represent the noise ceiling (e.g. inter-subject correlation in each ROI). High inter-subject correlation means low variance. Right panel: The Spearman’s correlation values are represented after normalization for the noise ceiling. Since the noise ceiling was obviously higher than the group mean correlation in both ROI, negative values were expected after normalization.

This information is, by itself, important. However, if we would normalize the results of the model comparisons according to these noise ceilings we would lose this information and we would even reverse the results of the correlation with the behavioural models (see Author response image 2, right side) and we would observe a higher correlation in VOTC than in the temporal lobe, which we believe would misleadingly suggest higher auditory categorical information in VOTC than in the temporal cortex if one did not consider the noise ceiling being much higher in the temporal cortex. As a side note, the negative values after the normalization were expected here, since in both ROIs the noise ceiling was higher compared to the group average correlation values.

We therefore think that the noise not only provide interesting information about the brain representations in the different groups but also, in complement to the correlation between the brain and model DSM, allows to evaluate fully and transparently the fine-grained structure of how a region codes the stimuli space. This is a point that we actually address more directly with the new inter-subjects analysis that we now ran to understand how variable is the brain representation in VOTC across subjects belonging either to the same group or to different groups (see new Figure 5 in the main manuscript for further details).

If we normalize the correlation between the representational similarity of VOTC and the behavioural model according to the noise ceiling of each group, we would indeed find a comparable amount of correlation in VOTC both for EBa and SCa (Author response image 3, right panel).

**Author response image 3. respfig3:** Dot-plot graphs representing the correlation of the behavioral model with the DSMs extracted from VOTC in both SCa (pink) and a EBa (green) groups. Each dot represents one subject, the colored (pink and green) lines represent the group mean values. The vertical black bars represent the standard error of the mean across subjects. Left panel: Spearman’s correlation with the behavioural model. The grey lines represent the noise ceiling (e.g. inter-subject correlation in each group). High inter-subject correlation means low variance. Right panel: The Spearman’s correlation values are represented after normalization for the noise ceiling. Since the noise ceiling was higher than the group mean correlation in both group, negative values were expected after normalization.

However, if we look at the data without the normalization (Author response image 3, left panel), we clearly see that there is a difference between the two groups, with the representation in SCa being much noisier and variable across subjects (therefore having a lower noise ceiling). We are convinced that, in case of normalization for noise ceiling, such absence of difference (following the reasoning we made above when comparing temporal and occipital regions) would be misleading.

For these reasons we think that the use of RSA, with the representation of both the correlation results and the noise ceiling (provided in each of our representation of RSA analyses) for each group and ROI is preferable for fully and transparently appreciating our results, also avoiding misleading the reader. Indeed, RSA provides a straightforward way of measuring the correspondence between the predicted (our models) and observed (the brain response) distances, in order to select the model that better explains the data in each group, given a certain level of noise.

In addition, there are also some methodological points that make RSA the best choice for our investigation (as highlighted by Diedrichsen and Kriegeskorte, 2017 in their work comparing PCM and RSA analyses). First of all, in case of a condition-rich design and for simple models such as it is the case in our study, RSA is much more computationally efficient compared than PCM. More importantly, the assumption of a linear relationship between predicted and measured representational dissimilarities (which is made by PCM), is not always a desirable choice for fMRI measurements since it might be violated in many cases (Driediechsen and Kriegeskorte, 2017). We now use the Spearman’s correlation to compare our models and the brain representations since this rank-correlation-based RSA provides a robust method without relying on a linear relationship between the predicted and measured dissimilarities (Driediechsen and Kriegeskorte, 2017).

Reviewer 3: The authors suggest that "VOTC reliably encodes sounds categories in blind people using a representational structure strikingly similar to the one found in vision". This conclusion is based on the magnitude of correlations between DSMs (e.g. Figure 3). However, I think it is important to go beyond the significant correlations and look carefully at the representational structure (e.g. see King et al., 2019). For example, while the dendrogram for SCv shows an initial split between the animate and inanimate items, for EBa the initial split is for the human conditions (HV, HN) from all others, and the animal conditions (AM, AM) and more closely grouped with BM and MG. Thus, the high correlation between SCv and EBa actually belies some differences in the representational structure that I think are worth commenting on. Similar comparison can be made for EVC and for SCa. So, I recommend the authors consider the representational structure in more detail and probably should draw back slightly from the claim that the representational structure in EBa and SCv are "strikingly similar".

We fully take this point and we followed the recommendation of the reviewers to go beyond the correlation values and to look more in details at the representational structure. This further exploration of our data is in line with the suggestion of toning down our claim about the similarity of the VOTC representational structure in EBa and SCv. In the Abstract, and elsewhere in the paper, we systematically substituted the wording “strikingly similar” with “partially similar”. More generally, we moved our attention from the similarity between the representational structure of our three groups towards highlighting some interesting between-groups differences.

To do so, we now use a hierarchical clustering approach similar to the one of King et al., 2019, as suggested by the reviewer. We describe the analysis in the new section of the Materials and Methods entitled Hierarchical clustering analysis on the brain categorical representations:

“In order to go beyond the correlation values and to explore more qualitatively the representational structure of VOTC and EVC in the 3 groups, we implemented a hierarchical clustering approach (King et al., 2019). […] We applied the same clustering analysis also on the behavioural data (see Figure 4—figure supplement 2)”.

Here are the results, reported in the paper under the section “Hierarchical clustering analysis on the brain categorical representation”:

“We implemented this analysis to go beyond the magnitude of correlation values and to qualitatively explore the representational structure of our ROIs in the 3 groups. […] In the EBa, instead, we find a different clustering structure with the animal and the human categories being separate“.

As now reported in the Discussion section, the results from the hierarchical clustering analysis, together with the Jaccard similarity analysis, highlight a domain-by-sensory experience interaction, with the animal category represented differently in blind compared to the sighted subjects.

As we now report in the Discussion:

“Previous studies already suggested that intrinsic characteristics of objects belonging to different categories might drive different representation in the VOTC of the blind (Wang et al., 2015; Bi et al., 2016).”

Compared to previous literature, our results highlight a further distinction within the animate category in early blind people. In this group, the animal category does not cluster together with the human stimuli both at behavioural and at brain level but tend to be assimilated to the inanimate categories. An explanation for this effect could be the different way blind and sighted individuals might have in perceiving and interacting with animals. In fact, if we exclude pets (only 1 out of the 6 animals we included in this study), sighted individuals normally perceive the animacy of animals (such as birds, donkey, horses etc.) mostly throughout vision (either in real life or in documentaries or videos). Blind people, instead, do normally learn the peculiar shape of each animal touching miniature, static, models of them. Moreover, when blind people hear the sounds of these animals without seeing them, they might combine these sounds with the rest of the environmental sounds, and this is indeed what we see in the behavioral ratings, in which only blind subjects cluster together animals and big environmental sounds.

2) One of the reviewers was particularly concerned about the usage of group means to characterise differences between groups, without adequately representing inter-subject variance within each group. In some analyses (e.g. the topographical selectivity map) the mean is taken as the sole group statistics (as a fixed effect), making the analysis highly susceptible to outliers and precluding generalisation to other samples. In other analyses (e.g. DSMs) a permutation test is included to determine group differences; however the specifics of this permutation tests are not well explained and require more thought – does this permutation test adequately captures the variance of the categorical representation between sighted and blind individuals? Or are the authors potentially confusing their units of analysis (see Parsons et al., eLife, 2018)? Ideally, variance should be assessed by measuring the effect in each individual participant and comparing the effect across groups (indeed this approach is used in some of the tests in the current study). In the case of DSM, each participant (e.g. from the EBa group) should be correlated to the SCv group, generating a distribution of the effect of interest. To an extent, a similar analysis is presented in Figure 5, providing much lower effect sizes than highlighted throughout the paper. So, we are looking for a more considered quantification of the main effects. A lot of confusion/misinterpretation could be avoided if the authors used standard statistics that takes into consideration the inter-subject variance, which is the key unit of your analysis (e.g. t-test and its variants). If the authors feel that their own approach is preferable, then they should increase the clarity on their analysis and its validity, with a clear statement of what null is.

We thank the reviewer for bringing this to our attention. We agree that in some of our statistical analyses we were not adequately considering the intersubject variance within each group. Following the recommendations made by the reviewers, we now implemented a different way of computing statistics. In general, the main point is that we do not use anymore, in any of the analyses, the group mean as a fixed effect, but we consider the variance measuring the effect in each individual participant.

To assess statistical differences, we now apply parametric tests (T-Test and ANOVA) in the analyses that met all the assumptions required by parametric statistics: normal distribution of the data, homogeneity of variances and independency of the observations. This was the case of the Jaccard similarity analyses (both within and between groups), of the voxels’ count analysis and of the averaged binary decoding analysis. However, in the rest of the cases we considered non-parametric statistic more appropriate since not meeting the criteria for parametric testing. Our choice of relying on non-parametric statistic (i.e. permutation analysis) is mainly driven by the most recent guidelines about the more appropriate way to run statistics on multivariate fMRI data (e.g. Stelzer et al., 2013). In this work the authors outline several theoretical points, supported by simulated data, according to which MVPA data might not respect some of the assumptions imposed by the t-test:

1) Normal distribution of the samples. One fundamental assumption of the t-test is that the samples need to be distributed normally, especially when the sample size is small. On the contrary, the unknown distribution of decoding accuracies is generally skewed and long-tailed and, in practice, depends massively on the classifier used and the input data itself.

2) Continuous distribution of the samples. A further assumption is that the underlying distribution from which samples are drawn should be continuous. This is not the case in accuracy values from decoding. In fact, the indicator function, which maps the number of correctly predicted labels to an accuracy value between 0 and 1, can only take certain values: for k cross-validation steps and a test set of size t, only k×t+1 different values between 0 and 1 can be taken.

3) Low variance of the samples is important for the t-test. Quite the opposite, the single subject accuracies are in general highly variable. In fact, the number of observations available for classification is very limited. This limitation of samples represents one of the main causes of the high variance of the accuracy values.

Similar guidelines concern also the correlation values between dissimilarity matrices (at the base of RSA analyses). In their seminal paper about how to implement RSA, Kriegeskorte and collaborators (2008) highlight that for dissimilarity matrices the independence of the samples cannot be assumed, because each similarity is dependent on two response patterns, each of which also codetermines the similarities of all its other pairings in the RDM. Therefore, they suggested to test the relatedness of dissimilarity matrices by using permutation tests (e.g. randomly permuting the conditions, to reorder rows and columns of one of the two dissimilarity matrices to be compared according to this permutation, and to compute the correlation).

That being said, we agree with the reviewers that in certain cases the use of standard statistics might avoid some confusion/misinterpretation and might be more straightforward. Therefore, we decided to move to parametric statistics for the analyses that allowed it (i.e. the analyses that met the main three assumptions required to apply the parametric statistics: normality of the distribution/ homogeneity of variances/ independency of data).

However, some of our analysis (i.e. the correlation of the topographical selectivity maps, the correlation of the brain dissimilarity matrices, the correlation of the RSA connectivity profiles and the inter-subject variance analysis) did not meet the assumption of independency of the data. This issue is related to the identification of the unit of analysis also mentioned by the reviewer in the current comment. We thank the reviewer for pointing out the interesting paper from Parsons et al., 2018, in which the problem of independency of the data is formally described. We now consider the guidance of this paper in building our statistical procedure.

Only in the experimental setting where each experimental unit provides a single outcome or observation, the experimental unit is the same as the unit of analysis (i.e. the one analyzed). In this case the independency of the data is maintained, and the use of parametric statistic is allowed. However, this is not always the case. For example, in our analyses previously mentioned (i.e. the correlation of the topographical selectivity maps, the correlation of the brain dissimilarity matrices, the correlation of the RSA connectivity profiles and the inter-subject variance analysis), our units of analysis are the correlation values between experimental units. In fact, in these analyses we contrast group’s comparisons, so data from the same subjects are always included in two comparisons (e.g. data from EBa subjects are included both in the SCv-EBa and in EBa-SCa comparisons). Therefore, the independency of the data is not respected here. In this case, using classical parametric statistics we would treat equally all observations in the analysis, ignoring the dependency in the data and this would lead to inflation of the false positive rate and incorrect (often highly inflated) estimates of statistical power, resulting in invalid statistical inference (Parsons et al., 2018).

For this reason, we believe that the use of permutation is a preferable approach in the case of these analyses. This approach is more flexible than the parametric statistics and allows to overcome the problem of independency. For example, in testing the difference between the comparisons we could build our null distribution keeping fixed the labels of the group common to the 2 pairs and shuffling the labels of the subjects from the other two groups (see next paragraphs for a more detailed description of all the statistical procedure used in the permutation analysis).

Also in the case of RSA with representational models we stick to permutation statistics, since also in this case there is a problem of independency of the data at the level of the DSMs’ building, as Kriegeskorte and collaborators highlighted in their seminal paper on RSA (2008): For dissimilarity matrices the independence of the samples cannot be assumed, because each similarity is dependent on two response patterns, each of which also codetermines the similarities of all its other pairings in the RDM. Therefore, the authors suggest testing the relatedness of dissimilarity matrices by using permutation tests (e.g. randomly permuting the conditions, to reorder rows and columns of one of the two dissimilarity matrices to be compared according to this permutation, and to compute the correlation).

In contrast to the previous version of our manuscript, we now changed the way of computing permutation, considering the variance of our values and we also explained more clearly our null hypothesis. In order to avoid confusion about the way we run statistics, we now explain all the information related to our statistical tests in a new section at the end of Materials and methods titled “Statistical analyses”:

“To assess statistical differences, we applied parametric tests (T-Test and ANOVA) in the analyses that met the main assumptions required by parametric statistics: normal distribution of the data and independency of the observations. […]

In each analysis, all the p-values are reported after false discovery rate (FDR) correction implemented using the matlab function ‘mafdr’”.

The main consequence of this change in the statistical methods is a decrease of the effect sizes in all the three analyses (correlation of the topographical selectivity maps; correlation of the brain dissimilarity matrices and correlation of the RSA connectivity profiles), as anticipated by the reviewer. However, even though the magnitude of the effect is reduced, this did not affect the significance of the results.

For example, if we look at the correlation between the brain DSMs, both in EVC and VOTC, we see that even though the average correlation between the DSMs in the new version of the results is lower compare to the previous results, however the same correlations that were significant in the previous results stay significant also in the new results (see Author response image 4).

**Author response image 4. respfig4:** New version of the results from the correlation of brain DSMs in EVC (left) and in VOTC (right). Despite a reduction of size effect in the new results, there is not any major change in the statistical results.

Similarly, the correlation values between the topographical maps remain significant (see Figure 1B).

Finally, a similar effect appears also in the correlation results between the RSA connectivity profiles (see Figure 8B). Also, in this case we find correlation values that are decreased in magnitude, but no difference in the significant tests:

3) A few of the reviewers were commenting on inconsistent statistical criteria, with the significance test flipping between one-tail to two-tails. For example, correlations between DSM were tested for significance using a one-tailed permutation test. This resulted in identifying a very strong negative correlation between the SCv and SCa in the early visual cortex (r = -.50), as non-significant, where in fact this effect is stronger than the correlation presented between SCv and EBa DSMs – r=.41, p=0.03). Similarly, the SCv DSM in the early visual cortex may be significantly negatively correlated to the behavioral similarity ratings (r=-0.09; the in sighted EVC DSM is reported to be significantly correlated with both the behavioural DSMs at r=.09, p=0.01). This could be an interesting finding, e.g. if cross-modal inhibition is categorically selective (allowing for decoding of sound categories in EVC in another study; (Vetter et al., 2014)). This finding could also add to the mechanistic explanation of the plasticity in the blind. But at present it is uninterpretable. We therefore request that one-tail testing would be avoided, unless there's strong statistical justification to usage it (e.g. in case of a replication of previous findings). In that case, the reasons for the one-tail hypothesis should be reported more transparently and interpreted more cautiously. But if the authors are confirming a new hypothesis, which we believe is the case for most of the tests reported, we'd encourage them to stick to two-tailed hypothesis testing. The authors are welcomed to consider correction for multiple comparisons, but again, we are looking for consistency throughout the manuscript and justification of the criteria used (or abandoned).

We thank the reviewer for raising this important point which give us the opportunity to clarify further our statistical strategy. We use one-tailed permutation test only in two analyses: the correlation of neural DSMs between groups and the correlation between neural DSMs and representational models (i.e. behavioural/low-level models); otherwise we systematically used 2-sided hypothesis testing. Our decision to use one-tail hypothesis testing in those specific condition is based on the lack of interpretability of negative correlation in RSA.

First, we would like to point out that the interesting possibility of a categorically selective cross-modal inhibition, suggested by the reviewer, would actually still produce a positive correlation with our behavioral/categorical model. In the use of RSA, in fact, we look at the structure of the representation of our categories in a specific brain region (i.e. how the representation of each category is similar or different from the representation of the other categories in a given ROI), however the structure of the representation does not tell anything about the level of activity of this region. For instance, we could see that in a specific brain ROI the representation of tools looks different from the representation of animals. From this information we cannot infer if the categorical representation embedded in the multivoxel pattern was the product of a deactivation or an activation, we can only infer that the brain activity patterns for tools are different from the brain activity patterns for animals. Therefore, a categorically selective cross-modal inhibition and a categorically selective cross-modal activation would create a similar structure of the representation, in both cases positively correlated with a categorical model.

A negative correlation with a representational model, such as the one we find in EVC of SCa with the behavioural model, highlight the fact that categories that look similar in our model have a different brain pattern activity while categories that look different in our model share a similar pattern of activity. These results are very difficult to interpret. One possibility is that there is another model, that we did not test, which is anti-correlated with our behavioural model and that can explain the representational structure of this ROI. This is one of the main limitations of RSA: there are in theory infinite models that we could test (Kriegeskorte et al., 2008). In practice, we need to select our model a priori for obvious statistical reasons (e.g. fitting a model a posteriori after observing the structure of our data).

Based on these limitations, studies relying on RSA typically do not interpret negative correlations’ results and tend to build unidirectional hypothesis testing (for an example with a design similar to the one of our study, including blind and sighted participants and using RSA to test different representational models, see Peelen et al. 2014; for recent examples of one-tailed test of RSA correlation see also (Fischer-Baum et al., 2017; Handjaras et al., 2017; Leshinskaya et al., 2017; Zhao et al., 2017; Wang et al., 2018; Evans et al., 2019).

In other words, since we cannot easily interpret negative correlations, it makes more sense to build a unidirectional hypothesis. We now added a short explanation in the paper to justify our choice of using a one-tailed instead than a two-tailed test:

”Considering the unidirectional hypothesis for this test (a positive correlation between neural similarity and models similarity) and the difficult interpretation of a negative correlation, one-tailed statistical tests were used. For all other tests (e.g., differences between groups), for which both directions might be hypothesized, two-tailed tests were (Peelen et al., 2014; Evans et al., 2019)”.

That being said, and in order to reassure the reviewers that using one-tailed instead of two-tailed test has no significant impact on the (positive) correlation results, we show the permutation results reporting the p values for both the one-tailed and the two-tailed test after FDR correction for the 12 multiple comparisons (see Author response image 5). As you can see, the only correlation values that are significant with the two-tailed and not with the one-tailed are the negative ones, that in any case (for the reasons previously reported) we would not have interpreted.

**Author response image 5. respfig5:** Comparison of the p-values (one-tailed vs two-tailed) resulting from the correlation of the brain DSMs (EVC on the left and VOTC on the right) with the representational behavioural and low-level models in the 3 groups (first line: SCv, middle line: EBa, lower line: SCa). The null distribution is represented in dark blue. The red line represents the actual correlation value. For each permutation test p-values are reported both for two-tailed and one-tailed tests. P value reported in red are significant according to the selected threshold of 0.05. p values are reported after FDR correction for 12 multiple comparisons.

Finally, the reviewer might have noticed that there is one difference compared to the results of this analysis in the previous version of the paper: the correlation with the behavioural model and the neural DSM from VOTC in SCa is not anymore significant. This is, however, unrelated to the use of one- or two-tailed test but instead related to the use of Spearman instead of Pearson correlation in our RSA analyses.

4) The focus on VOTC, and the specific definition of the ROIs requires further consideration. In particular the reviewers mentioned the lack of discrimination within it, limited coverage of the ventral stream, and lack of consideration of other areas which might be involved in categorical representation, e.g. high order auditory areas or other visual areas in the lateral occipital cortex (see also comment 6 below). Further exploratory analysis, focused by previous research on similar topics, would strengthen the interoperability of the results. A searchlight approach might be particularly helpful, though we leave it to the authors to decide on the specific methodology.

We thank the reviewer for raising this important point that leads us to clarify further our hypothesis-driven analytical strategy. There are three aspects that need to be clarified: 1) Why we decided to take the entire VOTC (and not separate subregions within it as region of interest; 2) How did we define our VOTC mask and why we chose that way; 3) Why we did not include other regions in our analyses.

First, since we were interested in the brain representation of different categories, we decided to focus on the ventro-occipito temporal cortex as a whole. This region is well known to contain several distinct macroscopic brain regions known to prefer a specific category of visual objects including faces, places, body parts, small artificial objects, etc. (Kanwisher, 2010). We decided to focus our analyses on a full mask of VOTC, and not in specific sub-parcels (such as FFA, PPA, etc.) because we were interested in looking at the categorical representation across categories and not at within a specific category. Our study therefore builds upon the paradigm shift of viewing VOTC as a distributed categorical system rather than a sum of isolated functionally specific areas, which reframes how we should expect to understand those areas (Haxby et al., 2001). In fact, our main aim was to investigate how input of presentation and visual experience impact on the general representation of different categories in the brain. Looking at one specific category selective region at a time would not allow us to address this specific question. This approach has already been validated by previous studies that investigated the categorical representation in the ventral-occipito temporal cortex using a wide VOTC mask (Kriegeskorte et al., 2008; Grill-Spector and Weiner, 2014; Wang et al., 2015; Xu et al., 2016; Hurk et al., 2017; Peelen and Downing, 2017; Ritchie et al., 2017).

That being said, our winner take all topographic analyses (see Figure 1B) in combination with calculation of Jaccard index (assessing topographic overlap) provide a clear measure of how each category maps onto a similar brain region within the VOTC mask across modalities and groups. This analyze suggest that there is a partial spatial overlap onto where separate categories map, but also highlight some differences (e.g. for animals).

Secondly, why did we decide to use a structural definition on VOTC mask and not a functional localizer? The main reason is that it is quite challenging to functionally localize VOTC in blind subjects functionally. Indeed, all the classical localizers are based on visual stimulation. Therefore, we thought that it would be misleading to functionally localize the mask only in the sighted and then apply it both to sighted and blind participants. In relation to the previous point, this is even more true if we had to localize specific categorically specific regions since we first need to localize those regions (e.g. FFA, PPA, LO) based on functional localizers since their locations show important inter-individual variability (Kanwisher, 2010; Julian et al., 2012). This is not feasible in blind people where clear functional localizers for those regions are not trivial to define. For this reason, we decided to rely on a structural definition of the mask. We decided to work in the subject space limiting the transformation and normalization of the brain’s images. This is particularly relevant when comparing blind and sighted subjects given that blindness is associated with significant changes in the structure of the brain itself, particularly within the occipital cortex (Pan et al., 2007; Jiang et al., 2009; Park et al., 2009; Dormal et al., 2016). That is why we used the anatomical scan to segment the brain in separate regions according to the Desikan-Killiany atlas (Desikan et al., 2006) implemented in FreeSurfer (http://surfer.nmr.mgh.harvard.edu). And finally, among the parcels produced by this atlas, we selected the regions laying within the ventral-occipito-temporal cortex and known to be involved in visual categorical representation of different categories.

Finally, we did not include other regions because our a priori hypothesis was based on VOTC. The exploration of additional parcels would be more based on an exploratory approach and would increase the multiple comparisons problem of our study. Because of this reason we decided to focus our analysis selectively on VOTC, adding the EVC as a control node. What drives the functional organization of VOTC (visual experience, retinotopy, curvature etc…) is a burgeoning topic with recent influential paper focusing precisely on such question (Hurk et la., 2017, Grill-Spector et al., 2014; Bracci et al., 2017;Bi et al., 2016; Peelen et al., 2017; Wang et al., 2015). We decided to insert our study in that topic by testing the role visual experience plays in driving the functional organization of VOTC by testing sighted and blind individuals with categorical sounds.

We agree with the reviewers that it would be interesting to investigate the categorical representation in other parts of the brain, such as the lateral occipital complex or the temporal cortex. We are indeed investigating at the moment the categorical representation in the temporal cortex of sighted and blind subjects (using a part of the actual dataset), with also an additional extension including blind subjects with late onset of blindness. However, we believe that this topic is beyond the scope of the present study that is already theoretically and technically challenging. Adding other regions would necessarily need changing the theoretical focus of the study and likely result in a paper presenting a patchwork results that are difficult to integrate in a streamlined global theoretical framework.

That being said, we agree with the reviewers that we needed to better explain our choice about the definition of our ROIs. We added some clarification in the section “Regions of interest”:

“Since we were interested in the brain representation of different categories we decided to focus on the ventro-occipito temporal cortex as a whole. […] Then, we combined these areas in order to obtain one bilateral EVC ROI and one bilateral VOTC ROI (Figure 3A). […]”

Finally, we thank the reviewer for highlighting the important point related to the searchlight approach. The searchlight approach might be, indeed, a useful and powerful analysis in the case of a more exploratory study. However, we again believe that this analysis is beyond the scope of the current paper, and we think that it might not help in adding value to our study which has a more hypothesis driven orientation (such as Wang et al., 2015; Hurk et al., 2017; Bi et al., 2016). On the opposite, the control for the (copious) multiple comparisons could even hinder part of the effects that emerge using the hypothesis-driven ROI approach. Moreover, as highlighted in the previous part of this comment, we decided to work in the subject space since there are plenty of evidences that early blindness is associated with significant changes in the structure of the brain itself, particularly within the occipital cortex (Dormal et al., 2016; Jiang et al., 2009; Pan et al., 2007; Park et al., 2009). Therefore, even though we could implement the searchlight approach in each individual subject, a normalization step would be required to move to a common space in order to infer the location at the group level. Working in individually defined anatomical masks allowed as to circumvent this problem.

In addition, even though we agree that the searchlight approach might be helpful as a side analysis to explore our data more extensively, we respectfully remind the reviewer that we build our neural dissimilarity matrices using a crossvalidation approach for each possible pair of conditions (i.e. for each ROI and for each group we run 28 binary decoding tests); we believe that it would be extremely challenging (especially for a side-analysis), at the computational level, to repeat the same analysis for each voxel.

As a final note, the multidimensional nature of fMRI data can always trigger new hypothesis-testing based on sensibilities of reviewers (which region is selected, which method to test the hypothesis etc.). We however believe that this can be problematic in generating post-hoc hypotheses and enlarging the theoretical and statistical space of a research project, potentially producing scientific malpractice (Zimring, Nature, 2016). More generally speaking, the request for additional experiments or analyses in the review phase has been recently eloquently debated by Hidde Ploegh in a Nature News called “End the Wasteful Tyranny of Reviewer Experiments” or by Derek Lowe in Science Translational Medicine: “Just A Few More Month’s Work, That’s All I’m Asking Here”. Quoting Ploegh here: “Submit a biomedical-research paper to Nature or other high-profile journals, and a common recommendation often comes back from reviewers: perform additional experiments. Although such extra work can provide important support for the results being presented, all too frequently it represents instead an entirely new phase of the project or does not extend the reach of what is reported”.

Again, our study was strongly hypothesis-driven by being focused on testing whether different modalities of input and sensory experiences affect the categorical representation in VOTC; following a focused analytical strategy as implemented by previous major literature in this field (Kriegeskorte et al., 2008; Grill-Spector et al., 2014; Bracci et al., 2017;Bi et al., 2016; Peelen et al., 2017; Wang et al., 2015).

5) The topographic selectivity analysis raised multiple comments from the reviewers. Beyond the issue raised in comment 2 above, it was agreed that this analysis potentially reveals important differences between the groups that are not adequately captured in the current presentation and analysis. For example, Reviewer 2 mentions: "the blind show nearly no preference for animal sounds in accordance with claims made by (Bi et al., 2016); in the sighted the human non-vocalizations are the preferred category for face selective areas in the visual cortex but in the auditory version (in both blind and sighted) there is mostly vocalization selectivity; The blind show little or no preference for the large-environmental sounds whereas the sighted show a differentiation between the two sound types". Reviewer 3 mentions: "In SCv, the medial ventral temporal cortex is primarily BIG-ENV, but in EBa, that same region is primarily BIG-MEC. Similarly, there appears to be more cortex showing VOC. In SCa than in either of the other two groups". We appreciate that some of the analysis might have been used as a replication of the Van den Hurk study, but we also expect them to stand alone, in terms of quality of analysis and interpretation. In particular, these topographical preference differences could be interesting and meaningful with relation to understanding blind visual cortex function, but it is hard to judge their strength or significance as the topographical selectivity maps are not thresholded, by activation strength or selectivity significance.

We thank the reviewers for raising these interesting comments about the topographic selectivity analysis. We agree that this analysis might reveal important differences between the groups and that we should try to adequately capture and highlight these interesting differences in our work.

We now compute the Jaccard index to quantify the similarity between the topographical maps preferentially elicited by each category in our 3 groups. The Jaccard similarity coefficient is a statistic used for measuring the topographic similarity and diversity of sample sets. The Jaccard coefficient is defined as the size of the intersection divided by the size of the union of the sample sets (see Author response image 6):

**Author response image 6. respfig6:** Visual representation of the intersection (left) and union (right) of two set of samples. The Jaccard coefficient is based on these two measures.

This value is 0 when the two sets are disjoint, 1 when they are equal, and between 0 and 1 otherwise. We used the Jaccard similarity index to look at two important aspects of our data: (1) the similarity of the topographical maps (for each category) between subjects from the same group (within group similarity). This analysis provides information about the consistency of the topographical representation of the categories across subjects within the same group (See Figure 1B-left). (2) The similarity between the topographical maps from both the blind and the sighted in the auditory experiment and the topographical map of the sighted in the visual experiment (between groups similarity). This analysis provides information about the similarity between the visual and auditory (both in sighted and in blind) topographical representations (see figure 1C).

For the within group similarity we computed the Jaccard similarity between the topographical selectivity map of each subject and the mean topographical map of his own group. This analysis produces 4 Jaccard similarity indices (one for each of the main category: (1) animal, (2) human, (3) manipulable objects and (4) big objects and places) for each group. Low values for one category mean that the topographical representation of that category varies a lot across subjects of the same group (e.g. the animal category in EBa). The results of the Jaccard similarity within each group are represented in the Figure 2B.

The results from the Jaccard similarity within groups are now reported in our article as follow:

“In order to look at the consistency of the topographical representation of the categories across subjects within the same group we computed the Jaccard similarity between the topographical selectivity map of each subject and the mean topographical map of his own group. […] The animals’ category showed a significantly lower Jaccard similarity within the EBa group compared to both SCa (p_FDR_ = 0.002) and SCv (p_FDR_ <0.001) groups while the similarity of big objects and places category was significantly higher in EBa compared to SCv (p_FDR_ = 0.038).”

For the between group analysis we computed the Jaccard similarity index for each category, between the topographical map of each blind and sighted subject in the auditory experiment and the averaged topographical selectivity map of the sighted in the visual experiment (see Figure 1C for the results). In more practical terms, this means that we have 4 Jaccard similarity indices (one for each of the main category: (1) animal, (2) human, (3) manipulable objects and (4) big objects and places) for each of the two group pairs: SCv-EBa and SCv-SCa.

The Figure 2C represents the results from this analysis.

We now present these new results from the Jaccard similarity between groups:

“In addition, we wanted to explore the similarity and differences among the topographical representations of our categories when they were presented visually compared to when they were presented acoustically, both in sighted and in blind. […] No difference emerged between groups, suggesting comparable level of similarity of the auditory topographical maps (both in blind and in sighted) with the visual topographical map in sighted participants.“

In addition to the Jaccard similarity analyses, we thought that a further relevant analysis would be to look at the number of voxels showing the preference for each category, in each group. In fact, the degree of overlap might be driven by the number of voxels selective for each category (e.g. it might be that there is no overlap for the animal category between SCv and EBa because almost no voxel prefer animal in blind).

The Figure 2B represents the number of selective voxels for each category and in every group.

We now present the results from the analysis on number of voxels selective for each category:

“Finally, since the degree of overlap highlighted by the Jaccard similarity values might be driven by the number of voxels selective for each category, we looked at the number of voxels showing the preference for each category in each group. […] The post hoc comparisons revealed that this difference was mainly driven by the reduced number of voxels selective for the animal category in EBa compared to SCv (p=0.02).”

We now include a new figure (Figure 2) in the new version of our manuscript in which we represent the results from these three analyses (i.e. number of selective voxels, Jaccard similarity within groups and Jaccard similarity with SCv).

We also added a section in the Discussion of our manuscript to integrate these results in the general framework of our paper and to relate them with previous studies. This section now reads as follows:

“Even though the categorical representation of VOTC appears, to a certain degree, immune to input modality and visual experience, there are also several differences emerging from the categorical representation of sight and sounds in the sighted and blind. […] Interestingly, in case of early visual deprivation this difference is also in the behavioural evaluation of the similarity of our stimuli.”

To go even more in details, in the supplemental material we included a representation of the same analyses applied on the 8 categories (see Figure 2—figure supplement 1).

The most striking observation from this analysis is that a large number of voxels show a preference for the big mechanical category (almost the double than the number of voxels preferring the other categories) in the VOTC of EBa group. In line with this result, the topographical map for the big mechanical objects is the most stable across EBa subjects, while the map for big environmental is the one less stable (i.e. lowest Jaccard similarity within EBa group). Moreover, also in this analysis, we see that in EBa a limited number of voxels is selective for both birds and mammals and the topographical maps of these two animals’ categories show a lower consistency across blind subjects.

Finally, it is true that our topographical selectivity maps are not thresholded by activation strength or selectivity significance. Our experiment was, definitely, designed to suit a multivariate fMRI analysis approach (e.g. fast event related design, short inter-stimulus interval, etc.) therefore our statistical power for univariate analyses might not be optimal. Moreover, it is important to keep in mind that we are looking at the activity of the occipital cortex for sounds, including in sighted individual, therefore we are not expecting high β-values in our univariate GLM. However, we believe that our topographical selectivity analysis is anyway valuable to make inference about some aspects of the categorical representation of the different categories across the different groups (for a similar approach see also Hurk et al., 2017; Striem-Amit et al., 2018). In other words, even though we cannot (and we do not) make any inference about the topographical category selectivity itself since our map are not thresholded, we however can use those maps for a second-order statistics (i.e. the correlation between the winner-take-all maps). It is important to highlight that if those maps were noisy, we would not observe any significant correlation. On the opposite, our results (supported by stringent statistics) show that the topographical map generated in VOTC by the auditory stimuli (both in blind and in sighted) is not stochastic; quite the opposite, it partially reflects the topographical map generated by the visual stimuli in sighted (see Figure 1B).

In addition, we would like some further clarification of how the correlations have been constructed, if indeed a use-takes-all approach is used to label each voxel.

To create the topographical selectivity map, we assign a label to each voxel within the VOTC mask. Because of the issue raised at point #2 concerning the usage of group mean to test statistical differences, we now compute a topographical selectivity map in each subject, in order to adequately represent inter-subject variance within each group. We improved the description of the analysis and the statistical tests in the “Materials and methods”, in the section related to the “*Topographical selectivity map”:*

“To create the topographical selectivity map (Figure 1B) we extracted in each participant the b-value for each of our 4 main conditions (animals, humans, manipulable objects and places) from each voxel inside the VOTC mask and we assigned to each voxel the condition producing the highest β-value (winner takes all). This analysis resulted in specific clusters of voxels that spatially distinguish themselves from their surround in terms of selectivity for a particular condition (Hurk et al., 2017, Striem-Amit et al., 2018).”

6) In the representational connectivity analysis, the authors may be overestimating the connectivity between regions due to contribution of intrinsic fluctuations between areas. To more accurately estimate the representational connectivity, it would be better if the authors used separate trials for the seed and target regions. See Henriksson et al., 2015, Neuroimage for a demonstration and discussion of this issue. Indeed, considering the clear differences found elsewhere between groups, the high between-group correlations are striking. It could have been informative to examine a control node (e.g. low-level visual cortex; low level auditory cortex) just to gain a better sense for what these correlations reveal.

We thank the reviewer for highlighting this point and for pointing out the relevant paper from Henriksson et al., 2015. In this paper, the authors reported that intrinsic cortical dynamics might strongly affect the representational geometry of a brain region, as reflected in response-pattern dissimilarities, and might exaggerate the similarity (quantified using a correlation measure) of representations between brain regions (Henriksson et al., 2015). In the paper the authors show that visual areas closer in cortex tended to exhibit greater similarity of representations. To bypass this problem, they suggest using independent data (e.g. data acquired in different runs) to compute the DSM in the seed ROI and the DSMs from the other brain ROIs that we want to correlate with the seed. This would, indeed, be a clever way to remove the intrinsic fluctuation bias.

However, we think that we are not under the influence of similar problems due to the way we built our dissimilarity matrices. As we are building our neural dissimilarity matrices using a cross-validation approach for each possible pair of conditions (and not 1 – the correlation value), we drastically reduce the possibility of this intrinsic fluctuation bias. As we explain in the Materials and methods section: “we preferred to use binary MVP-classification as dissimilarity index to build neural DSMs rather than other types of dissimilarity measures […] because when using a decoding approach, due to the intrinsic cross-validation steps, we would find that the two conditions that don’t drive responses are indistinguishable, despite their substantial correlation (Walther et al., 2016) since the noise is independent between the training and testing partitions, therefore cross-validated estimates of the distance do not grow with increasing noise. This was crucial in our study since we are looking at brain activity elicited by sounds in brain regions that are primarily visual (EVC and VOTC) where the SNR is expected to be low, at least in sighted people”.

In other words, the use of cross-validation across imaging runs ensures that the estimated distances between neural patterns are not systematically biased by run-specific noise (Whalther et al., 2016; Evans et al., 2019). Therefore, we believe that in our case the method suggested by Henriksson et al. is not necessary.

Moreover, the results we obtained from the representational connectivity analysis in VOTC already suggest that this intrinsic fluctuation bias cannot explain our results. If we look at the representational connectivity profiles represented in Figure 6, we clearly see that the proximity of the ROIs in the cortex is not systematically linked with a higher representational similarity (in contrast with what Henriksson et al. showed in their data). One example is the DSM form the fusiform node in that shows a higher similarity with the DSM from the inferior parietal cortex compared to the cuneus and even the lingual gyrus, which are nevertheless much closer in cortex.

Following the suggestion of the reviewers, we performed the same analysis also in the EVC as a control node, in order to gain a better sense for what these correlations reveal. In this ROI, we found a significant correlation only between the representational connectivity profiles of the two groups of sighted (SCv and SCa) and not between neither the EBa and the SCv nor between the EBa and the SCa. This highlight that the conclusion drawn from VOTC are specific and not the byproduct of a methodological property of how we compute our representational connectivity analysis. As we highlight in the Discussion:

“In support of this possibility, the representational connectivity profile of EVC in EBa did not show any similarity with the one of sighted (neither SCv nor SCa), suggesting a different way of crossmodal plasticity expression in this brain region”.

These results (together with other evidences coming from the decoding analysis and the RSA analysis) support the hypothesis that the posterior part of the occipital cortex in EB is the region that distance itself the most from the native computation it typically implements (Bi, Wang, and Caramazza, 2016; Büchel, 2003; Wang et al., 2015).

Importantly for the sake of this comment, from a methodological point of view, this is in support of the idea that the intrinsic cortical dynamics alone cannot explain our results, since the correlation between the representational connectivity profiles is sometimes absent.

Also, in this section it is not clear why the authors use a separate ROI for each of the 3 VOTC sub-regions, rather than the combination of all three, as they have for the main analysis? As a minor point, the reviewer wasn't clear how the authors switched from 3 nodes for each group to only one comparison for each group pair.

Our decision of using a separate ROI for each of the 3 VOTC sub-regions is mostly driven by previous literature. Previous studies have shown that, in sighted people, specific region within VOTC are functionally and/or structurally connected with specific extrinsic brain regions. For instance, the visual word form area (VWFA) in VOTC shows robust and specific anatomical connectivity to EVC and to frontotemporal language networks (Saygin et al., 2016). Similarly, the fusiform face area (FFA) shows a specific connectivity profile with other occipital regions (Saygin et al., 2012), and also a direct structural connection with the temporal voice area (TVA) in the superior temporal sulcus (Blank et al., 2011; Benetti et al., 2018) thought to support similar computations applied on faces and voices as well as their integration (Von Kriegstein, Kleinschmidt, Sterzer, and Giraud, 2005). In order to explore these large-scale brain connections in our 3 groups, we thought that it would be better to keep the specificity of the representational connectivity profile of each sub-region of VOTC. In this way we did not cancel out the differences across the representational connectivity profile of each VOTC sub-region and we ended-up with a higher variance in the connectivity profile of each subject, which is methodologically good for the following correlation analysis between the connectivity profiles. It is important to understand that we did not compare the connectivity profile of the single sub-regions, but in line with the rest of the paper, we concatenated the connectivity profile obtained from the three ROIs: our main aim was to keep more granularity in our connectivity profile.

However, to reassure the reviewers that keeping the entire VOTC as seed ROI would have a major impact on the results, we report the results from this analysis (Author response image 7, bottom panel), together with the same analysis using the 3 seeds ROIs (Author response image 7, top panel). As you can see the correlation values do not change critically in the two analyses.

**Author response image 7. respfig7:** Comparisons of the results from the RSA connectivity analysis using the 3 sub-regions of VOTC (fusiform, parahippocampal, infero-temporal) as seeds ROIs or VOTC as a single ROI.

Moreover, since the reviewers highlighted that our explanation on how we switched from 3 nodes for each group to only one comparison for each group pair was not clear enough, we now clarified this section in the new version of our manuscript:

”Finally, we computed the Spearman’s correlation between the 3 seed ROIs (i.e. fusiform gyrus, parahippocampal gyrus and infero-temporal cortex) and all the other 27 ROIs. We ended up with a connectivity profile of 3 (number of seeds) by 27 (ROIs representing the rest of the brain) for each subject. We considered this 3*27 matrix as one representational connectivity profile of the seed region (e.g. VOTC) in each subject.”